# Learning Deep Features in Instrumental Variable Regression

**Liyuan Xu**
Gatsby Unit
liyuan.jo.19@ucl.ac.uk

**Yutian Chen**
DeepMind
yutianc@google.com

**Siddarth Srinivasan**
University of Washington
sidsrini@cs.washington.edu

**Nando de Freitas**
DeepMind
nandodefreitas@google.com

**Arnaud Doucet**
DeepMind
arnauddoucet@google.com

**Arthur Gretton**
Gatsby Unit
arthur.gretton@gmail.com

## Abstract

Instrumental variable (IV) regression is a standard strategy for learning causal relationships between confounded treatment and outcome variables from observational data by using an instrumental variable, which affects the outcome only through the treatment. In classical IV regression, learning proceeds in two stages: stage 1 performs linear regression from the instrument to the treatment; and stage 2 performs linear regression from the treatment to the outcome, conditioned on the instrument. We propose a novel method, *deep feature instrumental variable regression (DFIV)*, to address the case where relations between instruments, treatments, and outcomes may be nonlinear. In this case, deep neural nets are trained to define informative nonlinear features on the instruments and treatments. We propose an alternating training regime for these features to ensure good end-to-end performance when composing stages 1 and 2, thus obtaining highly flexible feature maps in a computationally efficient manner. DFIV outperforms recent state-of-the-art methods on challenging IV benchmarks, including settings involving high dimensional image data. DFIV also exhibits competitive performance in off-policy policy evaluation for reinforcement learning, which can be understood as an IV regression task.

## 1 Introduction

The aim of supervised learning is to obtain a model based on samples observed from some data generating process, and to then make predictions about new samples generated from the same distribution. If our goal is to predict the effect of our actions on the world, however, our aim becomes to assess the influence of interventions on this data generating process. To answer such causal questions, a supervised learning approach is inappropriate, since our interventions, called *treatments*, may affect the underlying distribution of the variable of interest, which is called the *outcome*.

To answer these counterfactual questions, we need to learn how treatment variables causally affect the distribution process of outcomes, which is expressed in a *structural function*. Learning a structural function from observational data (that is, data where we can observe, but not intervene) is known to be challenging if there exists an unmeasured confounder, which influences both treatment and outcome. To illustrate: suppose we are interested in predicting sales of airplane tickets given price. During the holiday season, we would observe the simultaneous increase in sales and prices. This does not mean that raising the price

*causes* the sales to increase. In this context, the time of the year is a confounder, since it affects both the sales and the prices, and we need to correct the bias caused by it.

One way of correcting such bias is via *instrumental variable* (IV) regression (Stock and Trebbi, 2003). Here, the structural function is learned using instrumental variables, which only affect the treatment directly but not the outcome. In the sales prediction scenario, we can use supply cost shifters as the instrumental variable since they only affect the price (Wright, 1928; Blundell et al., 2012). Instrumental variables can be found in many contexts, and IV regression is extensively used by economists and epidemiologists. For example, IV regression is used for measuring the effect of a drug in the scenario of imperfect compliance (Angrist et al., 1996), or the influence of military service on lifetime earnings (Angrist, 1990). In this work, we propose a novel IV regression method, which can discover non-linear causal relationships using deep neural networks.

Classically, IV regression is solved by the *two-stage least squares* (2SLS) algorithm; we learn a mapping from the instrument to the treatment in the first stage, and learn the structural function in the second stage as the mapping from the conditional expectation of the treatment given the instrument (obtained from stage 1) to the outcome. Originally, 2SLS assumes linear relationships in both stages, but this has been recently extended to non-linear settings.

One approach has been to use non-linear feature maps. Sieve IV performs regression using a dictionary of nonlinear basis functions, which increases in size as the number of samples increases (Newey and Powell, 2003; Blundell et al., 2007; Chen and Pouzo, 2012; Chen and Christensen, 2018). Kernel IV (KIV) (Singh et al., 2019) and Dual IV regression (Muandet et al., 2020) use different (and potentially infinite) dictionaries of basis functions from reproducing kernel Hibert spaces (RKHS). Although these methods enjoy desirable theoretical properties, the flexibility of the model is limited, since all existing work uses pre-specified features.

Another approach is to perform the stage 1 regression through conditional density estimation (Carrasco et al., 2007; Darolles et al., 2011; Hartford et al., 2017). One advantage of this approach is that it allows for flexible models, including deep neural nets, as proposed in the DeepIV algorithm of (Hartford et al., 2017). It is known, however, that conditional density estimation is costly and often suffers from high variance when the treatment is high-dimensional.

More recently, Bennett et al. (2019) have proposed DeepGMM, a method inspired by the optimally weighted Generalized Method of Moments (GMM) (Hansen, 1982) to find a structural function ensuring that the regression residual and the instrument are independent. Although this approach can handle high-dimensional treatment variables and deep NNs as feature extractors, the learning procedure might not be as stable as the 2SLS approach, since it involves solving a smooth zero-sum game, as when training Generative Adversarial Networks (Wiatrak et al., 2019).

In this paper, we propose *Deep Feature Instrumental Variable Regression* (DFIV), which aims to combine the advantages of all previous approaches, while avoiding their limitations. In DFIV, we use deep neural nets to adaptively learn feature maps in the 2SLS approach, which allows us to fit highly nonlinear structural functions, as in DeepGMM and DeepIV. Unlike DeepIV, DFIV does not rely on conditional density estimation. Like sieve IV and KIV, DFIV learns the conditional expectation of the feature maps in stage 1 and uses the predicted features in stage 2, but with the additional advantage of learned features. We empirically show that DFIV performs better than other methods on several IV benchmarks, and apply DFIV successfully to off-policy policy evaluation, which is a fundamental problem in Reinforcement Learning (RL).

The paper is structured as follows. In Section 2, we formulate the IV regression problem and introduce two-stage least-squares regression. In Section 3, we give a detailed description of our DFIV method. We demonstrate the empirical performance of DFIV in Section 4, covering three settings: a classical demand prediction example from econometrics, a challenging IV setting where the treatment consists of high-dimensional image data, and the problem of off-policy policy evaluation in reinforcement learning.

## 2 PRELIMINARIES

### 2.1 PROBLEM SETTING OF INSTRUMENTAL VARIABLE REGRESSION

We begin with a description of the IV setting. We observe a treatment $X \in \mathcal{X}$, where $\mathcal{X} \subset \mathbb{R}^{d_X}$, and an outcome $Y \in \mathcal{Y}$, where $\mathcal{Y} \subset \mathbb{R}$. We also have an unobserved confounder that affects both $X$ and $Y$. This causal relationship can be represented with the following structural causal model:

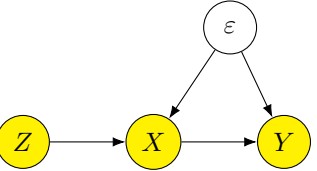

$$Y = f_{\text{struct}}(X) + \varepsilon, \quad \mathbb{E}\left[\varepsilon\right] = 0, \quad \mathbb{E}\left[\varepsilon|X\right] \neq 0, \quad (1)$$

Figure 1: Causal Graph.

where $f_{\text{struct}}$ is called the structural function, which we assume to be continuous, and $\varepsilon$ is an additive noise term. This specific confounding assumption is necessary for the IV problem. In Bareinboim and Pearl (2012), it is shown that we cannot learn $f_{\text{struct}}$ if we allow any type of confounders. The challenge is that $\mathbb{E}\left[\varepsilon|X\right] \neq 0$, which reflects the existence of a confounder. Hence, we cannot use ordinary supervised learning techniques since $f_{\text{struct}}(x) \neq \mathbb{E}\left[Y|X = x\right]$. Here, we assume there is no observable confounder but we may easily include this, as discussed in Appendix C.

To deal with the hidden confounder $\varepsilon$, we assume to have access to an instrumental variable $Z \in \mathcal{Z}$ which satisfies the following assumption.

**Assumption 1.** *The conditional distribution $P(X|Z)$ is not constant in $Z$ and $\mathbb{E}\left[\varepsilon|Z\right] = 0$.*

Intuitively, Assumption 1 means that the instrument $Z$ induces variation in the treatment $X$ but is uncorrelated with the hidden confounder $\varepsilon$. Again, for simplicity, we assume $\mathcal{Z} \subset \mathbb{R}^{d_Z}$. The causal graph describing these relationships is shown in Figure 1.[1]. Note that the instrument $Z$ cannot have an incoming edge from the latent confounder that is also a parent of the outcome.

Given Assumption 1, we can see that the function $f_{\text{struct}}$ satisfies the operator equation $\mathbb{E}\left[Y|Z\right] = \mathbb{E}\left[f_{\text{struct}}(X)|Z\right]$ by taking expectation conditional on $Z$ of both sides of (1). Newey and Powell (2003) provide necessary and sufficient conditions, known as completeness assumptions, to ensure identifiability of $f_{\text{struct}}(X)$. Solving this equation, however, is known to be ill-posed (Nashed and Wahba, 1974). To address this, recent works (Carrasco et al., 2007; Darolles et al., 2011; Muandet et al., 2020; Singh et al., 2019) minimize the following regularized loss $\mathcal{L}$ to obtain the estimate $\hat{f}_{\text{struct}}$:

$$\hat{f}_{\text{struct}} = \underset{f \in \mathcal{F}}{\arg\min} \, \mathcal{L}(f), \quad \mathcal{L}(f) = \mathbb{E}_{YZ}\left[(Y - \mathbb{E}_{X|Z}\left[f(X)\right])^2\right] + \Omega(f), \quad (2)$$

where $\mathcal{F}$ is an arbitrary space of continuous functions and $\Omega(f)$ is a regularizer on $f$.

### 2.2 TWO STAGE LEAST SQUARES REGRESSION

A number of works (Newey and Powell, 2003; Singh et al., 2019) tackle the minimization problem (2) using two-stage least squares (2SLS) regression, in which the structural function is modeled as $f_{\text{struct}}(x) = \boldsymbol{u}^\top \boldsymbol{\psi}(x)$, where $\boldsymbol{u}$ is a learnable weight vector and $\boldsymbol{\psi}(x)$ is a vector of fixed basis functions. For example, linear 2SLS used the identity map $\boldsymbol{\psi}(x) = x$, while sieve IV (Newey and Powell, 2003) uses Hermite polynomials.

In the 2SLS approach, an estimate $\hat{\boldsymbol{u}}$ is obtained by solving two regression problems successively. In stage 1, we estimate the conditional expectation $\mathbb{E}_{X|z}\left[\boldsymbol{\psi}(X)\right]$ as a function of $z$. Then in stage 2, as $\mathbb{E}_{X|z}\left[f(X)\right] = \boldsymbol{u}^\top \mathbb{E}_{X|z}\left[\boldsymbol{\psi}(X)\right]$, we minimize $\mathcal{L}$ with $\mathbb{E}_{X|z}\left[f(X)\right]$ being replaced by the estimate obtained in stage 1.

Specifically, we model the conditional expectation as $\mathbb{E}_{X|z}\left[\boldsymbol{\psi}(X)\right] = \boldsymbol{V}\boldsymbol{\phi}(z)$, where $\boldsymbol{\phi}(z)$ is another vector of basis functions and $\boldsymbol{V}$ is a *matrix* to be learned. Again, there exist many

---

[1]We show the simplest causal graph in Figure 1 It entails $Z \perp\!\!\!\perp \varepsilon$, but we only require $Z$ and $\varepsilon$ to be uncorrelated in Assumption 1. Of course, this graph also says that $Z$ is not independent of $\varepsilon$ when conditioned on observations $X$.

choices for $\phi(z)$, which can be infinite-dimensional, but we assume the dimensions of $\psi(x)$ and $\phi(z)$ to be $d_1, d_2 < \infty$ respectively.

In stage 1, the matrix $V$ is learned by minimizing the following loss,

$$\hat{V} = \underset{V \in \mathbb{R}^{d_1 \times d_2}}{\arg\min} \mathcal{L}_1(V), \quad \mathcal{L}_1(V) = \mathbb{E}_{X,Z}\left[\|\psi(X) - V\phi(Z)\|^2\right] + \lambda_1 \|V\|^2, \tag{3}$$

where $\lambda_1 > 0$ is a regularization parameter. This is a linear ridge regression problem with multiple targets, which can be solved analytically. In stage 2, given $\hat{V}$, we can obtain $u$ by minimizing the loss

$$\hat{u} = \underset{u \in \mathbb{R}^{d_1}}{\arg\min} \mathcal{L}_2(u), \quad \mathcal{L}_2(u) = \mathbb{E}_{Y,Z}\left[\|Y - u^\top \hat{V}\phi(Z)\|^2\right] + \lambda_2 \|u\|^2, \tag{4}$$

where $\lambda_2 > 0$ is another regularization parameter. Stage 2 corresponds to a ridge linear regression from $\hat{V}\phi(Z)$ to $Y$, and also enjoys a closed-form solution. Given the learned weights $\hat{u}$, the estimated structural function is $\hat{f}_{\text{struct}}(x) = \hat{u}^\top \psi(x)$.

## 3   DFIV ALGORITHM

In this section, we develop the DFIV algorithm. Similarly to Singh et al. (2019), we assume that we do not necessarily have access to observations from the joint distribution of $(X, Y, Z)$. Instead, we are given $m$ observations of $(X, Z)$ for stage 1 and $n$ observations of $(Y, Z)$ for stage 2. We denote the stage 1 observations as $(x_i, z_i)$ and the stage 2 observations as $(\tilde{y}_i, \tilde{z}_i)$. If observations of $(X, Y, Z)$ are given for both stages, we can evaluate the out-of-sample losses, and these losses can be used for hyper-parameter tuning of $\lambda_1, \lambda_2$ (Appendix A).

DFIV uses the following models

$$f_{\text{struct}}(x) = u^\top \psi_{\theta_X}(x) \quad \text{and} \quad \mathbb{E}_{X|z}[\psi_{\theta_X}(X)] = V\phi_{\theta_Z}(z), \tag{5}$$

where $u \in \mathbb{R}^{d_1}$ and $V \in \mathbb{R}^{d_1 \times d_2}$ are the parameters, and $\psi_{\theta_X}(x) \in \mathbb{R}^{d_1}$ and $\phi_{\theta_Z}(z) \in \mathbb{R}^{d_2}$ are the neural nets parameterised by $\theta_X \in \Theta_X$ and $\theta_Z \in \Theta_Z$, respectively. As in the original 2SLS algorithm, we learn $\mathbb{E}_{X|z}[\psi_{\theta_X}(X)]$ in stage 1 and $f_{\text{struct}}(x)$ in stage 2. In addition to the weights $u$ and $V$, however, we also learn the parameters of the feature maps, $\theta_X$ and $\theta_Z$. Hence, we need to alternate between stages 1 and 2, since the conditional expectation $\mathbb{E}_{X|z}[\psi_{\theta_X}(X)]$ changes during training.

**Stage 1 Regression**  The goal of stage 1 is to estimate the conditional expectation $\mathbb{E}_{X|z}[\psi_{\theta_X}(X)] \simeq V\phi_{\hat{\theta}_Z}(z)$ by learning the matrix $V$ and parameter $\theta_Z$, with $\theta_X = \hat{\theta}_X$ given and fixed. Given the stage 1 data $(x_i, z_i)$, this can be done by minimizing the empirical estimate of $\mathcal{L}_1$,

$$\hat{V}^{(m)}, \hat{\theta}_Z = \underset{V \in \mathbb{R}^{d_1 \times d_2}, \theta_Z \in \Theta_Z}{\arg\min} \mathcal{L}_1^{(m)}(V, \theta_Z), \ \mathcal{L}_1^{(m)} = \frac{1}{m}\sum_{i=1}^m \|\psi_{\hat{\theta}_X}(x_i) - V\phi_{\theta_Z}(z_i)\|^2 + \lambda_1 \|V\|^2. \tag{6}$$

Note that the feature map $\psi_{\hat{\theta}_X}(X)$ is fixed during stage 1, since this is the "target variable." If we fix $\theta_Z$, the minimization problem (6) reduces to a linear ridge regression problem with multiple targets, whose solution as a function of $\theta_X$ and $\theta_Z$ is given analytically by

$$\hat{V}^{(m)}(\theta_X, \theta_Z) = \Psi_1^\top \Phi_1 (\Phi_1^\top \Phi_1 + m\lambda_1 I)^{-1}, \tag{7}$$

where $\Phi_1, \Psi_1$ are feature matrices defined as $\Psi_1 = [\psi_{\theta_X}(x_1), \ldots, \psi_{\theta_X}(x_m)]^\top \in \mathbb{R}^{m \times d_1}$ and $\Phi_1 = [\phi_{\theta_Z}(z_1), \ldots, \phi_{\theta_Z}(z_m)]^\top \in \mathbb{R}^{m \times d_2}$. We can then learn the parameters $\theta_Z$ of the adaptive features $\psi_{\theta_Z}$ by minimizing the loss $\mathcal{L}_1^{(m)}$ at $V = \hat{V}^{(m)}(\hat{\theta}_X, \theta_Z)$ using gradient descent. For simplicity, we introduce a small abuse of notation by denoting as $\hat{\theta}_Z$ the result of a user-chosen number of gradient descent steps on the loss (6) with $\hat{V}^{(m)}(\hat{\theta}_X, \theta_Z)$ from (7), even though $\hat{\theta}_Z$ need not attain the minimum of the non-convex loss (6). We then write

$\hat{\boldsymbol{V}}^{(m)} := \hat{\boldsymbol{V}}^{(m)}(\hat{\theta}_X, \hat{\theta}_Z)$. While this trick of using an analytical estimate of the linear output weights of a deep neural network might not lead to significant gains in standard supervised learning, it turns out to be very important in the development of our 2SLS algorithm. As shown in the following section, the analytical estimate $\hat{\boldsymbol{V}}^{(m)}(\theta_X, \hat{\theta}_Z)$ (now considered as a function of $\theta_X$) will be used to backpropagate to $\theta_X$ in stage 2.

**Stage 2 Regression**   In stage 2, we learn the structural function by computing the weight vector $\boldsymbol{u}$ and parameter $\theta_X$ while *fixing* $\theta_Z = \hat{\theta}_Z$, and thus the corresponding feature map $\boldsymbol{\phi}_{\hat{\theta}_Z}(z)$. Given the data $(\tilde{y}_i, \tilde{z}_i)$, we can minimize the empirical version of $\mathcal{L}_2$, defined as

$$\hat{\boldsymbol{u}}^{(n)}, \hat{\theta}_X = \underset{\boldsymbol{u} \in \mathbb{R}^{d_1}, \theta_X \in \Theta_X}{\arg\min} \mathcal{L}_2^{(n)}(\boldsymbol{u}, \theta_X), \quad \mathcal{L}_2^{(n)} = \frac{1}{n} \sum_{i=1}^{n} (\tilde{y}_i - \boldsymbol{u}^\top \hat{\boldsymbol{V}}^{(m)} \boldsymbol{\phi}_{\hat{\theta}_Z}(\tilde{z}_i))^2 + \lambda_2 \|\boldsymbol{u}\|^2. \quad (8)$$

Again, for a given $\theta_X$, we can solve the minimization problem (8) for $\boldsymbol{u}$ as a function of $\hat{\boldsymbol{V}}^{(m)} := \hat{\boldsymbol{V}}^{(m)}(\theta_X, \hat{\theta}_Z)$ by a linear ridge regression

$$\hat{\boldsymbol{u}}^{(n)}(\theta_X, \hat{\theta}_Z) = \left( \hat{\boldsymbol{V}}^{(m)} \Phi_2^\top \Phi_2 (\hat{\boldsymbol{V}}^{(m)})^\top + n\lambda_2 I \right)^{-1} \hat{\boldsymbol{V}}^{(m)} \Phi_2^\top \boldsymbol{y}_2, \quad (9)$$

where $\Phi_2 = [\boldsymbol{\phi}_{\hat{\theta}_Z}(\tilde{z}_1), \dots, \boldsymbol{\phi}_{\hat{\theta}_Z}(\tilde{z}_n)]^\top \in \mathbb{R}^{n \times d_2}$ and $\boldsymbol{y}_2 = [\tilde{y}_1, \dots, \tilde{y}_n]^\top \in \mathbb{R}^n$.

The loss $\mathcal{L}_2^{(n)}$ explicitly depends on the parameters $\theta_X$ and we can backpropagate it to $\theta_X$ via $\hat{\boldsymbol{V}}^{(m)}(\theta_X, \hat{\theta}_Z)$, even though the samples of the treatment variable $X$ do not appear in stage 2 regression. We again introduce a small abuse of notation for simplicity, and denote by $\hat{\theta}_X$ the estimate obtained after a few gradient steps on (8) with $\hat{\boldsymbol{u}}^{(n)}(\theta_X, \hat{\theta}_Z)$ from (9), even though $\hat{\theta}_X$ need not minimize the non-convex loss (8). We then have $\hat{\boldsymbol{u}}^{(n)} = \hat{\boldsymbol{u}}^{(n)}(\hat{\theta}_X, \hat{\theta}_Z)$. After updating $\hat{\theta}_X$, we need to update $\hat{\theta}_Z$ accordingly. We do not attempt to backpropagate through the estimate $\hat{\theta}_Z$ to do this, however, as this would be too computationally expensive; instead, we alternate stages 1 and 2. We also considered updating $\hat{\theta}_X$ and $\hat{\theta}_Z$ jointly to optimize the loss $\mathcal{L}_2^{(n)}$, but this fails, as discussed in Appendix F.

**Computational Complexity and Convergence**   The computational complexity of the algorithm is $O(md_1 d_2 + d_2^3)$ for stage 1, while stage 2 requires additional $O(nd_1 d_2 + d_1^3)$ computations. This is small compared to KIV (Singh et al., 2019), which takes $O(m^3)$ and $O(n^3)$, respectively. We can further speed up the learning by using mini-batch training as shown in Algorithm 1. Here, $\hat{\boldsymbol{V}}^{(m_b)}$ and $\hat{\boldsymbol{u}}^{(n_b)}$ are the functions given by (7) and (9) calculated using mini-batches of data. Similarly, $\mathcal{L}_1^{(m_b)}$ and $\mathcal{L}_1^{(n_b)}$ are the stage 1 and 2 losses for the mini-batches. We recommend setting the batch size large enough so that $\hat{\boldsymbol{V}}^{(m_b)}, \hat{\boldsymbol{u}}^{(n_b)}$ do not diverge from $\hat{\boldsymbol{V}}^{(m)}, \hat{\boldsymbol{u}}^{(n)}$ computed on the entire dataset. Furthermore, we observe that setting $T_1 > T_2$, i.e. updating $\theta_Z$ more frequently than $\theta_X$, stabilizes the learning process.

In Appendix B, we provide regularity conditions under which the function learned by DFIV converges to the true structural function in probability. The derivation is based on Rademacher complexity bounds (Mohri et al., 2012).

## 4   EXPERIMENTS

In this section, we report the empirical performance of the DFIV method. The evaluation considers both low and high-dimensional treatment variables. We used the demand design dataset of Hartford et al. (2017) for benchmarking in the low and high-dimensional cases, and we propose a new setting for the high-dimensional case based on the dSprites dataset (Matthey et al., 2017). In the deep RL context, we also apply DFIV to perform off-policy policy evaluation (OPE). The network architecture and hyper-parameters are provided in Appendix G. The algorithms in the first two experiments are implemented using PyTorch (Paszke et al., 2019) and the OPE experiments are implemented using TensorFlow (Abadi et al., 2015) and the Acme RL framework (Hoffman et al., 2020). The code is included in the supplemental material.

---

**Algorithm 1** Deep Feature Instrumental Variable Regression

---

**Input:** Stage 1 data $(x_i, z_i)$, Stage 2 data $(\tilde{y}_i \tilde{z}_i)$, Regularization parameters $(\lambda_1, \lambda_2)$. Initial values $\hat{\theta}_X, \hat{\theta}_Z$. Mini-batch size $(m_b, n_b)$. Number of updates in each stage $(T_1, T_2)$.
**Output:** Estimated structural function $\hat{f}_{\text{struct}}(x)$
 1: **repeat**
 2:    Sample $m_b$ stage 1 data $(x_i^{(b)}, z_i^{(b)})$ and $n_b$ stage 2 data $(\tilde{y}_i^{(b)}, \tilde{z}_i^{(b)})$.
 3:    **for** $t = 1$ to $T_1$ **do**
 4:        Return function $\hat{\boldsymbol{V}}^{(m_b)}(\hat{\theta}_X, \theta_Z)$ in (7) using $(x_i^{(b)}, z_i^{(b)})$
 5:        Update $\hat{\theta}_Z \leftarrow \hat{\theta}_Z - \alpha \nabla_{\theta_Z} \mathcal{L}_1^{(m_b)}(\hat{\boldsymbol{V}}^{(m_b)}(\hat{\theta}_X, \theta_Z), \theta_Z)|_{\theta_Z = \hat{\theta}_Z}$      \\ *Stage 1 learning*
 6:    **end for**
 7:    **for** $t = 1$ to $T_2$ **do**
 8:        Return function $\hat{\boldsymbol{u}}^{(n_b)}(\theta_X, \hat{\theta}_Z)$ in (9) using $(\tilde{y}_i^{(b)}, \tilde{z}_i^{(b)})$ and function $\hat{\boldsymbol{V}}^{(m_b)}(\theta_X, \hat{\theta}_Z)$
 9:        Update $\hat{\theta}_X \leftarrow \hat{\theta}_X - \alpha \nabla_{\theta_X} \mathcal{L}_2^{(n_b)}(\hat{\boldsymbol{u}}^{(n_b)}(\theta_X, \hat{\theta}_Z), \theta_X)|_{\theta_X = \hat{\theta}_X}$      \\ *Stage 2 learning*
10:    **end for**
11: **until convergence**
12: Compute $\hat{\boldsymbol{u}}^{(n)} := \hat{\boldsymbol{u}}^{(n)}(\hat{\theta}_X, \hat{\theta}_Z)$ from (9) using entire dataset.
13: **return** $\hat{f}_{\text{struct}}(x) = (\hat{\boldsymbol{u}}^{(n)})^\top \boldsymbol{\psi}_{\hat{\theta}_X}(x)$

---

## 4.1 DEMAND DESIGN EXPERIMENTS

The demand design dataset is a synthetic dataset introduced by Hartford et al. (2017) that is now a standard benchmarking dataset for testing nonlinear IV methods. In this dataset, we aim to predict the demands on airplane tickets $Y$ given the price of the tickets $P$. The dataset contains two observable confounders, which are the time of year $T \in [0, 10]$ and customer groups $S \in \{1, ..., 7\}$ that are categorized by the levels of price sensitivity. Further, the noise in $Y$ and $P$ is correlated, which indicates the existence of an unobserved confounder. The strength of the correlation is represented by $\rho \in [0, 1]$. To correct the bias caused by this hidden confounder, the fuel price $C$ is introduced as an instrumental variable. Details of the data generation process can be found in Appendix E.1. In DFIV notation, the treatment is $X = P$, the instrument is $Z = C$, and $(T, S)$ are the observable confounders.

We compare the DFIV method to three leading modern competitors, namely KIV (Singh et al., 2019), DeepIV (Hartford et al., 2017), and DeepGMM (Bennett et al., 2019). We used the DFIV method with observable confounders, as introduced in Appendix C. Note that DeepGMM does not have an explicit mechanism for incorporating observable confounders. The solution we use, proposed by Bennett et al. (2019, p. 2), is to incorporate these observables in both instrument *and* treatment; hence we apply DeepGMM with treatment $X = (P, T, S)$ and instrumental variable $Z = (C, T, S)$. Although this approach is theoretically sound, this makes the problem unnecessary difficult since it ignores the fact that we only need to consider the conditional expectation of $P$ given $Z$.

We used a network with a similar number of parameters to DeepIV as the feature maps in DFIV and models in DeepGMM. We tuned the regularizers $\lambda_1, \lambda_2$ as discussed in Appendix A, with the data evenly split for stage 1 and stage 2. We varied the correlation parameter $\rho$ and dataset size, and ran 20 simulations for each setting. Results are summarized in Figure 2. We also evaluated the performance via the estimation of average treatment effect and conditional average treatment effect, which is presented in Appendix E.2

Next, we consider a case, introduced by Hartford et al. (2017), where the customer type $S \in \{1, \ldots, 7\}$ is replaced with an image of the corresponding handwritten digit from the MNIST dataset (LeCun and Cortes, 2010). This reflects the fact that we cannot know the exact customer type, and thus we need to estimate it from noisy high-dimensional data. Note that although the confounder is high-dimensional, the treatment variable is still real-valued, i.e. the price $P$ of the tickets. Figure 3 presents the results for this high-dimensional confounding case. Again, we train the networks with a similar number of learnable parameters to DeepIV in DFIV and DeepGMM, and hyper-parameters are set in the way discussed in

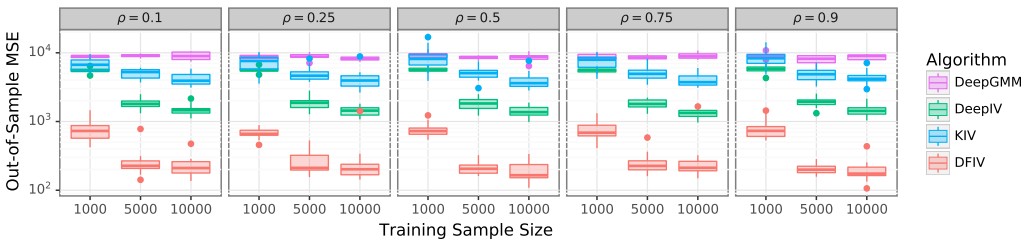

Figure 2: MSE for demand design dataset with low dimensional confounders.

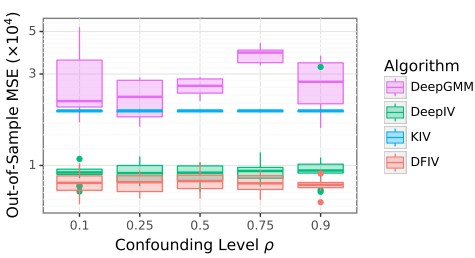

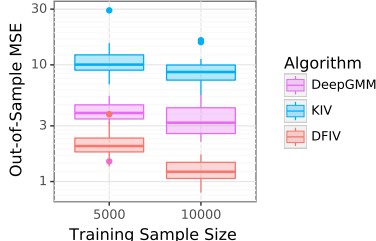

Figure 3: MSE for demand design dataset with high dimensional observed confounders.

Figure 4: MSE for dSprite dataset. DeepIV did not yield meaningful predictions for this experiment.

Appendix A. We ran 20 simulations with data size $n + m = 5000$ and report the mean and standard error.

Our first observation from Figure 2 and 3 is that the level $\rho$ of correlation has no significant impact on the error under any of the IV methods, indicating that all approaches correctly account for the effect of the hidden confounder. This is consistent with earlier results on this dataset using DeepIV and KIV (Hartford et al., 2017; Singh et al., 2019). We note that DeepGMM does not perform well in this demand design problem. This may be due to the current DeepGMM approach to handling observable confounders, which might not be optimal. KIV performed reasonably well for small sample sizes and low-dimensional data, but it did less well in the high-dimensional MNIST case due to its less expressive features. In high dimensions, DeepIV performed well, since the treatment variable is unidimensional. However, DFIV performed consistently better than all other methods in both low and high dimensions, which suggests it can learn a flexible structural function in a stable manner.

## 4.2 DSPRITES EXPERIMENTS

To test the performance of DFIV methods for a high dimensional treatment variable, we utilized the dSprites dataset (Matthey et al., 2017). This is an image dataset described by five latent parameters (`shape, scale, rotation, posX` and `posY`). The images are $64 \times 64 = 4096$-dimensional. In this experiment, we fixed the `shape` parameter to `heart`, i.e. we only used heart-shaped images. An example is shown in Figure 5.

From this dataset, we generated data for IV regression in which we use each figure as treatment variable $X$. Hence, the treatment variable is 4096-dimensional in this experiment. To make the task more challenging, we used `posY` as the hidden confounder, which is not revealed to the model. We used the other three latent variables as the instrument variables $Z$. The structural function $f_{\text{struct}}$ and outcome $Y$ are defined as

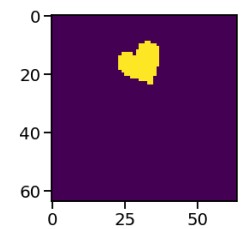

Figure 5: dSprite image

$$f_{\text{struct}}(X) = \frac{\|AX\|_2^2 - 5000}{1000}, \quad Y = f_{\text{struct}}(X) + 32(\texttt{posY} - 0.5) + \varepsilon, \quad \varepsilon \sim \mathcal{N}(0, 0.5),$$

where each element of the matrix $A \in \mathbb{R}^{10 \times 4096}$ is generated from $\mathrm{Unif}(0.0, 1.0)$ and fixed throughout the experiment. See Appendix E.3 for the detailed data generation process.

We tested the performance of DFIV with KIV and DeepGMM, where the hyper-parameters are determined as in the demand design problem. The results are displayed in Figure 4. DFIV consistently yields the best performance of all the methods. DeepIV is not included in the figure because it fails to give meaningful predictions due to the difficulty of performing conditional density estimation for the high-dimensional treatment variable. The performance of KIV suffers since it lacks the feature richness to express a high-dimensional complex structural function. Although DeepGMM performs comparatively to DFIV, we observe some instability during the training, see Appendix E.4.

## 4.3 Off-Policy Policy Evaluation Experiments

We apply our IV methods to the off-policy policy evaluation (OPE) problem (Sutton and Barto, 2018), which is one of the fundamental problems of deep RL. In particular, it has been realized by Bradtke and Barto (1996) that 2SLS could be used to estimate a linearly parameterized value function, and we use this reasoning as the basis of our approach. Let us consider the RL environment $\langle \mathcal{S}, \mathcal{A}, P, R, \rho_0, \gamma \rangle$, where $\mathcal{S}$ is the state space, $\mathcal{A}$ is the action space, $P : \mathcal{S} \times \mathcal{A} \times \mathcal{S} \to [0, 1]$ is the transition function, $R : \mathcal{S} \times \mathcal{A} \times \mathcal{S} \times \mathbb{R} \to \mathbb{R}$ is the reward distribution, $\rho_0 : \mathcal{S} \to [0, 1]$ is the initial state distribution, and discount factor $\gamma \in (0, 1]$. Let $\pi$ be a policy, and we denote $\pi(a|s)$ as the probability of selecting action $a$ in stage $s \in \mathcal{S}$. Given policy $\pi$, the $Q$-function is defined as

$$Q^\pi(s, a) = \mathbb{E}\left[\sum_{t=0}^{\infty} \gamma^t r_t \middle| s_0 = s, a_0 = a\right]$$

with $a_t \sim \pi(\cdot \mid s_t), s_{t+1} \sim P(\cdot|s_t, a_t), r_t \sim R(\cdot|s_t, a_t, s_{t+1})$. The goal of OPE is to evaluate the expectation of the $Q$-function with respect to the initial state distribution for a given target policy $\pi$, $\mathbb{E}_{s \sim \rho_0, a|s \sim \pi}[Q^\pi(s, a)]$, learned from a fixed dataset of transitions $(s, a, r, s')$, where $s$ and $a$ are sampled from some potentially unknown distribution $\mu$ and behavioral policy $\pi_b(\cdot|s)$ respectively. Using the Bellman equation satisfied by $Q^\pi$, we obtain a structural causal model of the form (1),

$$r = \overbrace{Q^\pi(s, a) - \gamma Q^\pi(s', a')}^{\text{structural function } f_{\text{struct}}(s, a, s', a')} \tag{10}$$
$$+ \underbrace{\gamma\left(Q^\pi(s', a') - \mathbb{E}_{s' \sim P(\cdot|s, a), a' \sim \pi(\cdot|s')}\left[Q^\pi(s', a')\right]\right) + r - \mathbb{E}_{r \sim R(\cdot|s, a, s')}[r]}_{\text{confounder } \varepsilon},$$

where $X = (s, a, s', a'), Z = (s, a), Y = r$. We have that $\mathbb{E}[\varepsilon] = 0, \mathbb{E}[\varepsilon|X] \neq 0$, and Assumption 1 is verified. Minimizing the loss (2) for the structural causal model (10) corresponds to minimizing the following loss $\mathcal{L}_{\text{OPE}}$

$$\mathcal{L}_{\text{OPE}} = \mathbb{E}_{s, a, r}\left[\left(r + \gamma \mathbb{E}_{s' \sim P(\cdot|s, a), a' \sim \pi(\cdot|s')}\left[Q^\pi(s', a')\right] - Q^\pi(s, a)\right)^2\right], \tag{11}$$

and we can apply any IV regression method to achieve this. In Appendix D, we show that minimizing $\mathcal{L}_{\text{OPE}}$ corresponds to minimizing the mean squared Bellman error (MSBE) (Sutton and Barto, 2018, p. 268) and we detail the DFIV algorithm for OPE. Note that MBSE is also the loss minimized by the residual gradient (RG) method proposed in (Baird, 1995) to estimate $Q$-functions. However, this method suffers from the "double-sample" issue, i.e. it requires two independent samples of $s'$ starting from the same $(s, a)$ due to the inner conditional expectation (Baird, 1995), whereas IV regression methods do not suffer from this issue.

We evaluate DFIV on three BSuite (Osband et al., 2019) tasks: catch, mountain car, and cartpole. See Section E.6.1 for a description of those tasks. The original system dynamics are deterministic. To create a stochastic environment, we randomly replace the agent action by a uniformly sampled action with probability $p \in [0, 0.5]$. The noise level $p$ controls the level of confounding effect. The target policy is trained using DQN (Mnih et al., 2015), and we subsequently generate an offline dataset for OPE by executing the policy in the

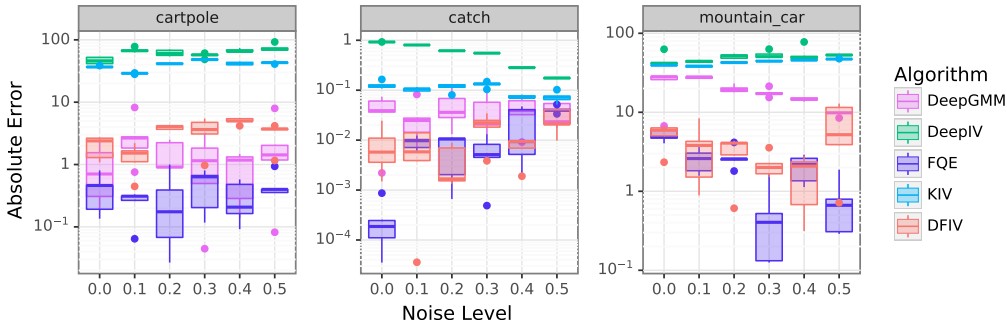

Figure 6: Error of offline policy evaluation.

same environment with a random action probability of 0.2 (on top of the environment's random action probability $p$). We compare DFIV with KIV, DeepIV, and DeepGMM; as well as Fitted Q Evaluation (FQE) (Le et al., 2019; Voloshin et al., 2019), a specialized approach designed for the OPE setting, which serves as our "gold standard" baseline (Paine et al., 2020) (see Section E.6.2 for details). All methods use the same network for value estimation. Figure 6 shows the absolute error of the estimated policy value by each method with a standard deviation from 5 runs. In catch and mountain car, DFIV comes closest in performance to FQE, and even matches it for some noise settings, whereas DeepGMM is somewhat worse in catch, and significantly worse in mountain car. In the case of cartpole, DeepGMM performs somewhat better than DFIV, although both are slightly worse than FQE. DeepIV and KIV both do poorly across all RL benchmarks.

## 5 CONCLUSION

We have proposed a novel method for instrumental variable regression, Deep Feature IV (DFIV), which performs two-stage least squares regression on flexible and expressive features of the instrument and treatment. As a contribution to the IV literature, we showed how to adaptively learn these feature maps with deep neural networks. We also showed that the off-policy policy evaluation (OPE) problem in deep RL can be interpreted as a nonlinear IV regression, and that DFIV performs competitively in this domain. This work thus brings the research worlds of deep offline RL and causality from observational data closer.

In terms of future work, it would be interesting to adapt the ideas from (Angrist and Krueger, 1995; Angrist et al., 1999; Hansen and Kozbur, 2014) to select the regularization hyperparameters of DFIV as well as investigate generalizations of DFIV beyond the additive model (1) as considered in (Carrasco et al., 2007, Section 5.5). In RL, problems with additional confounders are common, see e.g. (Namkoong et al., 2020; Shang et al., 2019), and we believe that adapting DFIV to this setting will be of great value.

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

## A HYPER-PARAMETER TUNING

If observations from the joint distribution of $(X, Y, Z)$ are available in both stages, we can tune the regularization parameters $\lambda_1, \lambda_2$ using the approach proposed in Singh et al. (2019). Let the complete data of stage 1 and stage 2 be denoted as $(x_i, y_i, z_i)$ and $(\tilde{x}_i, \tilde{y}_i, \tilde{z}_i)$. Then, we can use the data not used in each stage to evaluate the out-of-sample performance of the other stage. Specifically, the regularizing parameters are given by

$$\lambda_1^* = \arg\min \mathcal{L}_{1\text{-oos}}^{(n)}, \quad \mathcal{L}_{1\text{-oos}}^{(n)} = \frac{1}{n}\sum_{j=1}^{n}\|\psi_{\hat{\theta}_X}(\tilde{x}_i) - \hat{V}^{(m)}\phi_{\hat{\theta}_Z}(\tilde{z}_i)\|^2,$$

$$\lambda_2^* = \arg\min \mathcal{L}_{2\text{-oos}}^{(m)}, \quad \mathcal{L}_{2\text{-oos}}^{(m)} = \frac{1}{m}\sum_{i=1}^{m}(y_i - (\hat{u}^{(n)})^\top \hat{V}^{(m)}\phi_{\hat{\theta}_Z}(z_i))^2,$$

where $\hat{u}^{(n)}, \hat{V}^{(m)}, \hat{\theta}_{\theta_X}, \hat{\theta}_{\theta_Z}$ are the parameters learned in (6) and (8).

## B CONSISTENCY OF DFIV

In this appendix, we prove consistency of the DFIV approach. Our contribution is to establish consistency of the end-to-end procedure incorporating Stages 1 and 2, which we achieve by first showing a Stage 1 consistency result (Lemma 1), and then establishing the consistency of Stage 2 when the empirical Stage 1 solution is used as input (Lemma 2). The desired result then follows in Theorem 1.

Consistency results will be expressed in terms of the complexity of the function classes used in Stages 1 and 2, as encoded in the Rademacher complexity of functionals of these functions (see Proposition 1 below). Consistency for particular function classes can then be shown by establishing that the respective Rademacher complexities vanish. We leave for future work the task of demonstrating this property for function classes of interest.

### B.1 OPERATOR VIEW OF DFIV

The goal of DFIV is to learn structural function $f_{\text{struct}}$, which satisfies

$$\mathbb{E}_{Y|Z}[Y] = \mathbb{E}_{X|Z}[f_{\text{struct}}(X)]. \tag{12}$$

We model $f_{\text{struct}}$ as $f_{\text{struct}}(X) = u^\top \psi_{\theta_X}(X)$ and denote the hypothesis spaces for $\psi_{\theta_X}$ and $f_{\text{struct}}$ as follows:

$$\mathcal{H}_\psi = \{\psi_{\theta_X} : \mathcal{X} \to \mathbb{R}^{d_1} \mid \theta_X \in \Theta_X\},$$
$$\mathcal{F} = \{u^\top \psi_{\theta_X} : \mathcal{X} \to \mathcal{Y} \mid \psi_{\theta_X} \in \mathcal{H}_\psi, u \in \mathbb{R}^{d_1}\}.$$

To learn the parameters, we minimize the following stage 2 loss:

$$\hat{u}^{(n)}, \hat{\theta}_X = \arg\min_{u \in \mathbb{R}^{d_1}, \theta_X} \mathcal{L}_2^{(n)}(u, \theta_X), \quad \mathcal{L}_2^{(n)} = \frac{1}{n}\sum_{i=1}^{n}(\tilde{y}_i - u^\top \hat{\mathbb{E}}_{X|Z}[\psi_{\theta_X}(X)](\tilde{z}_i))^2.$$

We denote the resulting estimated structural function as $\hat{f}_{\text{struct}}(X) = (\hat{u}^{(n)})^\top \psi_{\hat{\theta}_X}(X)$. For simplicity, we set all regularization terms to zero. Here, $\hat{\mathbb{E}}_{X|Z}[\psi_{\theta_X}(X)]$ is the empirical conditional expectation operator, which maps an element of $\mathcal{H}_\psi$ to some function $g : \mathcal{Z} \mapsto \mathbb{R}^{d_1} \in \mathcal{G}$ that

$$\hat{\mathbb{E}}_{X|Z}[\psi_{\theta_X}(X)] = \arg\min_{g \in \mathcal{G}} \mathcal{L}_1^{(m)}(V, \theta_Z), \ \mathcal{L}_1^{(m)} = \frac{1}{m}\sum_{i=1}^{m}\|\psi_{\theta_X}(x_i) - g(z_i)\|^2.$$

In DFIV, we define $\mathcal{G}$ as

$$\mathcal{G} = \{V\phi_{\theta_Z} : \mathcal{Z} \to \mathbb{R}^{d_1} \mid V \in \mathbb{R}^{d_2 \times d_1}, \theta_Z \in \Theta_Z\}.$$

This formulation is equivalent to the one introduced in Section 3. With a slight abuse of notation, for $f(X) = u^\top \psi_{\theta_X}(X) \in \mathcal{F}$, we define $\hat{\mathbb{E}}_{X|Z}[f(X)]$ to be

$$\hat{\mathbb{E}}_{X|Z}[f(X)] = u^\top \hat{\mathbb{E}}_{X|Z}[\psi_{\theta_X}(X)]$$

since this is the empirical estimate of $\mathbb{E}_{X|Z}[f(X)]$.

### B.2 GENERALIZATION ERRORS FOR REGRESSION

Here, we bound the generalization errors of both stages using Rademacher complexity bounds (Mohri et al., 2012).

**Proposition 1.** *(Theorem 3.3 Mohri et al., 2012, with slight modification) Let $\mathcal{S}$ be a measurable space and $\mathcal{H}$ be a family of functions mapping from $\mathcal{S}$ to $[0, M]$. Given fixed dataset $S = (s_1, s_2, \ldots, s_n) \in \mathcal{S}^n$, empirical Rademacher complexity is given by*

$$\hat{\mathfrak{R}}_S(\mathcal{H}) = \mathbb{E}_{\boldsymbol{\sigma}} \left[ \sup_{h \in \mathcal{H}} \sum_{i=1}^n \sigma_i h(s_i) \right],$$

*where $\boldsymbol{\sigma} = (\sigma_1, \ldots, \sigma_n)$, with $\sigma_i$ is independent uniform random variables taking values in $\{-1, +1\}$. Then, for any $\delta > 0$, with probability at least $1 - \delta$ over the draw of an i.i.d sample $S$ of size $n$, each of following holds for all $h \in \mathcal{H}$:*

$$\mathbb{E}[h(s)] \leq \frac{1}{n} \sum_{i=1}^n h(s_i) + 2\hat{\mathfrak{R}}_S + 3M\sqrt{\frac{\log 2/\delta}{2n}},$$

$$\frac{1}{n} \sum_{i=1}^n h(s_i) \leq \mathbb{E}[h(s)] + 2\hat{\mathfrak{R}}_S + 3M\sqrt{\frac{\log 2/\delta}{2n}}.$$

We list the assumptions below.

**Assumption 2.** *The following hold:*

1. *Bounded outcome variable $|Y| \leq M$.*

2. *Bounded stage 1 hypothesis space: $\forall z \in \mathcal{Z}, \|\boldsymbol{g}(z)\| \leq 1$.*

3. *Bounded stage 2 feature map $\forall x \in \mathcal{X}, \|\boldsymbol{\psi}_{\theta_X}(x)\| \leq 1$.*

4. *Bounded stage 2 weight: $\|\boldsymbol{u}\| \leq M$.*

5. *Identifiable stage 1 hypothesis space: $\forall \boldsymbol{\psi}_{\theta_X} \in \mathcal{H}_{\boldsymbol{\psi}}, \exists g \in \mathcal{G}, \mathbb{E}_{X|Z}[\boldsymbol{\psi}_{\theta_X}] = g$.*

6. *Identifiable stage 2 hypothesis space: $f_{\text{struct}} \in \mathcal{F}$.*

Note that we leave aside questions of optimization. Thus, we assume that the optimization procedure over $(\theta_Z, \boldsymbol{V})$ is sufficient to recover $\hat{\mathbb{E}}_{X|Z}$, and that the optimization procedure over $(\boldsymbol{u}, \theta_X)$ is sufficient to recover $\hat{f}_{\text{struct}}$ (which requires, in turn, the correct $\hat{\mathbb{E}}_{X|Z}$, for this $\boldsymbol{\psi}_{\theta_X}$). We emphasize that Algorithm 1 does not guarantee these properties.

Based on these assumptions, we derive the generalization error in terms of $L_2$-norm $\|\cdot\|_{P(Z)}$ defined as

$$\|h\|_{P(Z)} = \left( \int \|h(Z)\|^2 \mathrm{d}P(Z) \right)^{\frac{1}{2}}.$$

The following lemma proves the generalization error for stage 1 regression.

**Lemma 1.** *Under Assumption 2, and given stage 1 data $S_1 = \{(x_i, z_i)\}_{i=1}^m$, for any $\delta > 0$, with at least probability $1 - 2\delta$, we have*

$$\left\| \hat{\mathbb{E}}_{X|Z}[f(X)] - \mathbb{E}_{X|Z}[f(X)] \right\|_{P(Z)} \leq M\sqrt{4\hat{\mathfrak{R}}_{S_1}(\mathcal{H}_1) + 24\sqrt{\frac{\log 2/\delta}{2m}}}$$

*for any $f = \boldsymbol{u}^\top \boldsymbol{\psi}_{\theta_X} \in \mathcal{F}$, where hypothesis space $\mathcal{H}_1$ is defined as*

$$\mathcal{H}_1 = \{(x, z) \in \mathcal{X} \times \mathcal{Z} \mapsto \|\boldsymbol{\psi}_{\theta_X}(x) - \boldsymbol{g}(z)\|^2 \in \mathbb{R} \mid \boldsymbol{g} \in \mathcal{G}, \boldsymbol{\psi}_{\theta_X} \in \mathcal{H}_{\boldsymbol{\psi}}\}. \tag{13}$$

*Proof.* From Cauchy–Schwarz inequality, we have

$$\left\| \mathbb{E}\left[f(X)|Z\right] - \hat{\mathbb{E}}_{X|Z}\left[f(X)\right] \right\|_{P(Z)} = \left\| \left( \boldsymbol{u}^\top \left( \mathbb{E}_{X|Z}\left[\boldsymbol{\psi}_{\theta_X}(X)\right] - \hat{\mathbb{E}}_{X|Z}\left[\boldsymbol{\psi}_{\theta_X}(X)\right] \right) \right) \right\|_{P(Z)}$$

$$\leq M \left\| \mathbb{E}_{X|Z}\left[\boldsymbol{\psi}_{\theta_X}(X)\right] - \hat{\mathbb{E}}_{X|Z}\left[\boldsymbol{\psi}_{\theta_X}(X)\right] \right\|_{P(Z)}$$

By applying Proposition 1 to hypothesis space $\mathcal{H}_1$, we have

$$\mathbb{E}_{XZ}\left[ \left\| \boldsymbol{\psi}_{\theta_X}(X) - \hat{\mathbb{E}}_{X|Z}\left[\boldsymbol{\psi}_{\theta_X}(X)\right] \right\|^2 \right]$$

$$\leq \frac{1}{m}\sum_{i=1}^{m} \left\| \boldsymbol{\psi}_{\theta_X}(x_i) - \hat{\mathbb{E}}_{X|Z}\left[\boldsymbol{\psi}_{\theta_X}(X)\right](z_i) \right\|^2 + 2\hat{\mathfrak{R}}_{S_1}(\mathcal{H}_1) + 12\sqrt{\frac{\log 2/\delta}{2m}}$$

with probability $1 - \delta$. Indeed all functions $h \in \mathcal{H}_1$ satisfy $\|h\| \leq 4$ since $\|\boldsymbol{\psi}_{\theta_X}(X)\| \leq 1, \|\boldsymbol{g}(Z)\| \leq 1$ from Assumption 2.

Also, since $\forall \boldsymbol{\psi}_{\theta_X} \in \mathcal{H}_{\boldsymbol{\psi}}, \exists g \in \mathcal{G}, \mathbb{E}_{X|Z}\left[\boldsymbol{\psi}_{\theta_X}\right] = g$, again from Proposition 1, we have

$$\frac{1}{m}\sum_{i=1}^{m} \left\| \boldsymbol{\psi}_{\theta_X}(x_i) - \mathbb{E}_{X|Z}\left[\boldsymbol{\psi}_{\theta_X}(X)\right](z_i) \right\|^2$$

$$\leq \mathbb{E}_{XZ}\left[ \left\| \boldsymbol{\psi}_{\theta_X}(X) - \mathbb{E}_{X|Z}\left[\boldsymbol{\psi}_{\theta_X}(X)\right] \right\|^2 \right] + 2\hat{\mathfrak{R}}_{S_1}(\mathcal{H}_1) + 12\sqrt{\frac{\log 2/\delta}{2m}}$$

with probability $1 - \delta$. From the optimality of $\hat{\mathbb{E}}_{X|Z}\left[\boldsymbol{\psi}_{\theta_X}(X)\right]$, we have

$$\frac{1}{m}\sum_{i=1}^{m} \left\| \boldsymbol{\psi}_{\theta_X}(x_i) - \hat{\mathbb{E}}_{X|Z}\left[\boldsymbol{\psi}_{\theta_X}\right](z_i) \right\|^2 \leq \frac{1}{m}\sum_{i=1}^{m} \left\| \boldsymbol{\psi}_{\theta_X}(x_i) - \mathbb{E}_{X|Z}\left[\boldsymbol{\psi}_{\theta_X}(X)\right](z_i) \right\|^2 .$$

Hence, we have

$$\mathbb{E}_{XZ}\left[ \left\| \boldsymbol{\psi}_{\theta_X}(X) - \hat{\mathbb{E}}_{X|Z}\left[\boldsymbol{\psi}_{\theta_X}(X)\right] \right\|^2 \right]$$

$$\leq \mathbb{E}_{XZ}\left[ \left\| \boldsymbol{\psi}_{\theta_X}(X) - \mathbb{E}_{X|Z}\left[\boldsymbol{\psi}_{\theta_X}(X)\right] \right\|^2 \right] + 4\hat{\mathfrak{R}}_{S_1}(\mathcal{H}_1) + 24\sqrt{\frac{\log 2/\delta}{2m}}$$

with probability $1 - 2\delta$. Now we have

$$\mathbb{E}_{XZ}\left[ \left\| \boldsymbol{\psi}_{\theta_X}(X) - \hat{\mathbb{E}}_{X|Z}\left[\boldsymbol{\psi}_{\theta_X}(X)\right] \right\|^2 \right]$$

$$= \mathbb{E}_{XZ}\left[ \left\| \boldsymbol{\psi}_{\theta_X}(X) - \mathbb{E}_{X|Z}\left[\boldsymbol{\psi}_{\theta_X}(X)\right] \right\|^2 \right] + \mathbb{E}_Z\left[ \left\| \mathbb{E}_{X|Z}\left[\boldsymbol{\psi}_{\theta_X}(X)\right] - \hat{\mathbb{E}}_{X|Z}\left[\boldsymbol{\psi}_{\theta_X}(X)\right] \right\|^2 \right],$$

and thus

$$\mathbb{E}_Z\left[ \left\| \mathbb{E}_{X|Z}\left[\boldsymbol{\psi}_{\theta_X}(X)\right] - \hat{\mathbb{E}}_{X|Z}\left[\boldsymbol{\psi}_{\theta_X}(X)\right] \right\|^2 \right] \leq 4\hat{\mathfrak{R}}_{S_1}(\mathcal{H}_1) + 24\sqrt{\frac{\log 2/\delta}{2m}} .$$

Therefore, by taking the square root of both sides, we can see

$$\left\| \hat{\mathbb{E}}_{X|Z}\left[f(X)\right] - \mathbb{E}_{X|Z}\left[f(X)\right] \right\|_{P(Z)} \leq M\sqrt{4\hat{\mathfrak{R}}_{S_1}(\mathcal{H}_1) + 24\sqrt{\frac{\log 2/\delta}{2m}}} .$$

$\square$

The generalization error for stage 2 is given in the following lemma.

**Lemma 2.** *Under Assumption 2, and given stage 1 data $S_1 = \{(x_i, z_i)\}_{i=1}^m$, stage 2 data $S_2 = \{(\tilde{y}_i, \tilde{z}_i)\}_{i=1}^n$, and the estimated structural function $\hat{f}_{\mathrm{struct}}(x) = (\hat{\boldsymbol{u}}^{(n)})^\top \boldsymbol{\psi}_{\hat{\boldsymbol{\theta}}_{\boldsymbol{x}}}(x)$, then for any $\delta > 0$, with at least probability $1 - 4\delta$, we have*

$$\left\| \mathbb{E}_{Y|Z}[Y] - \hat{\mathbb{E}}_{X|Z}\left[\hat{f}_{\mathrm{struct}}(X)\right] \right\|_{P(Z)}$$

$$\leq \sqrt{4\hat{\mathfrak{R}}_{S_2}(\mathcal{H}_2) + 24M^2\sqrt{\frac{\log 2/\delta}{2n}}} + M\sqrt{4\hat{\mathfrak{R}}_{S_1}(\mathcal{H}_1) + 24\sqrt{\frac{\log 2/\delta}{2m}}},$$

*where $\mathcal{H}_1$ is defined in (13) and $\mathcal{H}_2$ is defined as*

$$\mathcal{H}_2 = \{(y, z) \in \mathcal{Y} \times \mathcal{Z} \mapsto (y - \boldsymbol{u}^\top \boldsymbol{g}(z))^2 \in \mathbb{R} \mid \boldsymbol{g} \in \mathcal{G}, \boldsymbol{u} \in \mathbb{R}^{d_1}, \|\boldsymbol{u}\| \leq M\}. \tag{14}$$

*Proof.* Since $\hat{\mathbb{E}}_{X|Z}\left[\boldsymbol{\psi}_{\hat{\boldsymbol{\theta}}_{\boldsymbol{x}}}\right] \in \mathcal{G}$, by applying Proposition 1 to hypothesis space $\mathcal{H}_2$, we have

$$\mathbb{E}_{YZ}\left[(Y - \hat{\mathbb{E}}_{X|Z}\left[\hat{f}_{\mathrm{struct}}(X)\right])^2\right]$$

$$\leq \frac{1}{n}\sum_{i=1}^n \left(\tilde{y}_i - \hat{\mathbb{E}}_{X|Z}\left[\hat{f}_{\mathrm{struct}}(X)\right](\tilde{z}_i)\right)^2 + 2\hat{\mathfrak{R}}_{S_2}(\mathcal{H}_2) + 12M^2\sqrt{\frac{\log 2/\delta}{2n}}$$

with probability of $1 - \delta$. Note that all functions $h \in \mathcal{H}_2$ are bounded in $\|h\| \leq 4M^2$ since $|Y| \leq M, \|\boldsymbol{\psi}_{\theta_X}(X)\| \leq 1, \|\boldsymbol{u}\| \leq M$ from Assumption 2. Similarly, since $f_{\mathrm{struct}} \in \mathcal{F}$ from Assumption 2, we have

$$\frac{1}{n}\sum_{i=1}^n \left(\tilde{y}_i - \hat{\mathbb{E}}_{X|Z}\left[f_{\mathrm{struct}}(X)\right](\tilde{z}_i)\right)^2$$

$$\leq \mathbb{E}_{YZ}\left[(Y - \hat{\mathbb{E}}_{X|Z}\left[f_{\mathrm{struct}}(X)\right])^2\right] + 2\hat{\mathfrak{R}}_{S_2}(\mathcal{H}_2) + 12M^2\sqrt{\frac{\log 2/\delta}{2n}}$$

with probability $1 - \delta$. From the minimality of $\hat{f}_{\mathrm{struct}}$, we have

$$\frac{1}{n}\sum_{i=1}^n \left(\tilde{y}_i - \hat{\mathbb{E}}_{X|Z}\left[\hat{f}_{\mathrm{struct}}(X)\right](\tilde{z}_i)\right)^2 \leq \frac{1}{n}\sum_{i=1}^n (\tilde{y}_i - \hat{\mathbb{E}}_{X|Z}\left[f_{\mathrm{struct}}(X)\right](\tilde{z}_i))^2$$

Hence, we have

$$\mathbb{E}_{YZ}\left[\left(Y - \hat{\mathbb{E}}_{X|Z}\left[\hat{f}_{\mathrm{struct}}(X)\right]\right)^2\right]$$

$$\leq \mathbb{E}_{YZ}\left[(Y - \hat{\mathbb{E}}_{X|Z}\left[f_{\mathrm{struct}}(X)\right])^2\right] + 4\hat{\mathfrak{R}}_{S_2}(\mathcal{H}_2) + 24M^2\sqrt{\frac{\log 2/\delta}{2n}},$$

with probability $1 - 2\delta$. Now we also have

$$\mathbb{E}_{YZ}\left[\left(Y - \hat{\mathbb{E}}_{X|Z}\left[\hat{f}_{\mathrm{struct}}\right]\right)^2\right] = \mathbb{E}_{YZ}\left[(Y - \mathbb{E}_{Y|Z}[Y])^2\right] + \mathbb{E}_Z\left[(\hat{\mathbb{E}}_{X|Z}\left[\hat{f}_{\mathrm{struct}}\right] - \mathbb{E}_{Y|Z}[Y])^2\right],$$

$$\mathbb{E}_{YZ}\left[(Y - \hat{\mathbb{E}}_{X|Z}\left[f_{\mathrm{struct}}\right])^2\right] = \mathbb{E}_{YZ}\left[(Y - \mathbb{E}_{Y|Z}[Y])^2\right] + \mathbb{E}_Z\left[(\hat{\mathbb{E}}_{X|Z}\left[f_{\mathrm{struct}}\right] - \mathbb{E}_{Y|Z}[Y])^2\right].$$

Therefore, with probability $1 - 2\delta$,

$$\mathbb{E}_Z\left[(\mathbb{E}_{Y|Z}[Y] - \hat{\mathbb{E}}_{X|Z}\left[\hat{f}_{\mathrm{struct}}(X)\right])^2\right]$$

$$\leq 4\hat{\mathfrak{R}}_{S_2}(\mathcal{H}_2) + 24M^2\sqrt{\frac{\log 2/\delta}{2n}} + \mathbb{E}_Z\left[(\hat{\mathbb{E}}_{X|Z}\left[f_{\mathrm{struct}}(X)\right] - \mathbb{E}_{Y|Z}[Y])^2\right].$$

From Lemma 1 and (12), with probability at least $1 - 2\delta$,

$$\mathbb{E}_Z\left[(\hat{\mathbb{E}}_{X|Z}\left[f_{\mathrm{struct}}(X)\right] - \mathbb{E}_{Y|Z}[Y])^2\right] \leq 4M^2\hat{\mathfrak{R}}_{S_1}(\mathcal{H}_1) + 24M^2\sqrt{\frac{\log 2/\delta}{2m}}.$$

By combining them, we can see that with probability $1 - 4\delta$,

$$\left\| \mathbb{E}_{Y|Z}[Y] - \hat{\mathbb{E}}_{X|Z}\left[\hat{f}_{\text{struct}}(X)\right] \right\|_{P(Z)}$$

$$\leq \sqrt{4\hat{\mathfrak{R}}_{S_2}(\mathcal{H}_2) + 24M^2\sqrt{\frac{\log 2/\delta}{2n}} + 4M^2\hat{\mathfrak{R}}_{S_1}(\mathcal{H}_1) + 24M^2\sqrt{\frac{\log 2/\delta}{2m}}}$$

$$\leq \sqrt{4\hat{\mathfrak{R}}_{S_2}(\mathcal{H}_2) + 24M^2\sqrt{\frac{\log 2/\delta}{2n}}} + M\sqrt{4\hat{\mathfrak{R}}_{S_1}(\mathcal{H}_1) + 24\sqrt{\frac{\log 2/\delta}{2m}}}.$$

$\square$

### B.3 Consistency Proof of DFIV

The goal is to bound the deviation between $\hat{f}_{\text{struct}}$ and $f_{\text{struct}}$. This discrepancy is measured by $L_2$-norm with respect to $P(X)$, defined as

$$\|h(X)\|_{P(X)} = \left(\int \|h(X)\|^2 \mathrm{d}P(X)\right)^{\frac{1}{2}}.$$

However, we used the norm $\|\cdot\|_{P(Z)}$ in Lemmas 1 and 2. To bridge this gap, we state the necessary condition for identification introduced in (Newey and Powell, 2003).

**Proposition 2.** *(Proposition 2.1 Newey and Powell, 2003) For all $\delta(x)$ with finite expectation, $\mathbb{E}[\delta(X)|Z] = 0$ a.e. on $Z$ implies $\delta(x) = 0$.*

Proposition 2 is the minimum requirement for identification and Assumption 1 is a sufficient condition for it. Given Proposition 2, we can consider a constant $\tau < \infty$ defined as

$$\tau = \sup_{f \in \mathcal{F}, f \neq f_{\text{struct}}} \frac{\|f_{\text{struct}}(X) - f(X)\|_{P(X)}}{\|\mathbb{E}[f_{\text{struct}}(X) - f(X)|Z]\|_{P(Z)}}. \tag{15}$$

Note that $\tau \geq 1$ from definition and $\tau = 1$ if and only if $X$ is measurable with respect to $Z$. This measures the ill-posedness of IV problem. A similar quantity is introduced in (Blundell et al., 2007). Given this quantity, we derive the convergence rate of DFIV as follows.

**Theorem 1.** *Let Assumption 2 hold. Given stage 1 data $S_1 = \{(x_i, z_i)\}_{i=1}^m$, stage 2 data $S_2 = \{(\tilde{y}_i, \tilde{z}_i)\}_{i=1}^m$ and $\tau$ defined in (15), for any $\delta > 0$, with at least probabilty of $1 - 6\delta$, we have*

$$\|f_{\text{struct}}(X) - \hat{f}_{\text{struct}}(X)\|_{P(X)}$$

$$\leq 2\tau M\sqrt{4\hat{\mathfrak{R}}_{S_1}(\mathcal{H}_1) + 24\sqrt{\frac{\log 2/\delta}{2m}}} + \tau\sqrt{4\hat{\mathfrak{R}}_{S_2}(\mathcal{H}_2) + 24M^2\sqrt{\frac{\log 2/\delta}{2n}}},$$

*where $\mathcal{H}_1$ and $\mathcal{H}_2$ are defined in (13) and (14), respectively.*

*Proof.* Using $\tau$ in (15), we have

$$\|f_{\text{struct}}(X) - \hat{f}_{\text{struct}}(X)\|_{P(X)} \leq \tau \left\| \mathbb{E}\left[f_{\text{struct}}(X) - \hat{f}_{\text{struct}}(X)\Big|Z\right] \right\|_{P(Z)}$$

$$\leq \tau \left\| \mathbb{E}\left[\hat{f}_{\text{struct}}(X)|Z\right] - \hat{\mathbb{E}}_{X|Z}\left[\hat{f}_{\text{struct}}(X)\right] \right\|_{P(Z)}$$

$$+ \tau \left\| \mathbb{E}\left[f_{\text{struct}}(X)|Z\right] - \hat{\mathbb{E}}_{X|Z}\left[\hat{f}_{\text{struct}}(X)\right] \right\|_{P(Z)}$$

$$= \tau \left\| \mathbb{E}\left[\hat{f}_{\text{struct}}(X)|Z\right] - \hat{\mathbb{E}}_{X|Z}\left[\hat{f}_{\text{struct}}(X)\right] \right\|_{P(Z)}$$

$$+ \tau \left\| \mathbb{E}[Y|Z] - \hat{\mathbb{E}}_{X|Z}\left[\hat{f}_{\text{struct}}(X)\right] \right\|_{P(Z)},$$

where the last inequality holds from (12). Using Lemmas 1 and 2, the result thus follows. $\square$

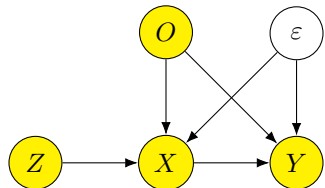

Figure 7: Causal graph with observable confounder

From this result, we obtain directly the following corollary.

**Corollary 1.** *Let Assumption 2 hold. If $\hat{\mathfrak{R}}_{S_1}(\mathcal{H}_1) \to 0$ and $\hat{\mathfrak{R}}_{S_2}(\mathcal{H}_2) \to 0$ in probability as datasize increases, $\hat{f}_{\text{struct}}$ converges to $f_{\text{struct}}$ in probability.*

The proof of vanishing Rademacher complexities for particular Stage 1 and Stage 2 function classes is a topic for future work.

## C  2SLS ALGORITHM WITH OBSERVABLE CONFOUNDERS

In this appendix, we formulate the DFIV method when observable confounders are available. Here, we consider the causal graph given in Figure 7. In addition to treatment $X \in \mathcal{X}$, outcome $Y \in \mathcal{Y}$, and instrument $Z \in \mathcal{Z}$, we have an observable confounder $O \in \mathcal{O}$. The structural function $f_{\text{struct}}$ we aim to learn is now $\mathcal{X} \times \mathcal{O} \to \mathcal{Y}$, and the structural causal model is represented as

$$Y = f_{\text{struct}}(X, O) + \varepsilon, \quad \mathbb{E}\left[\varepsilon\right] = 0, \quad \mathbb{E}\left[\varepsilon | X\right] \neq 0.$$

For hidden confounders, we rely on Assumption 1. For observable confounders, we introduce a similar assumption.

**Assumption 3.** *The conditional distribution $P(X|Z, O)$ is not constant in $Z, O$ and $\mathbb{E}\left[\varepsilon | Z, O\right] = 0$.*

Following a similar reasoning as in Section 2, we can estimate the structural function $\hat{f}_{\text{struct}}$ by minimizing the following loss:

$$\hat{f}_{\text{struct}} = \underset{f \in \mathcal{F}}{\arg\min}\, \tilde{\mathcal{L}}(f), \quad \tilde{\mathcal{L}}(f) = \mathbb{E}_{YZO}\left[(Y - \mathbb{E}_{X|Z,O}\left[f(X, O)\right])^2\right] + \Omega(f).$$

One universal way to deal with the observable confounder is to augment both the treatment and instrumental variables. Let us introduce the new treatment $\tilde{X} = (X, O)$ and instrument $\tilde{Z} = (Z, O)$, then the loss $\tilde{\mathcal{L}}$ becomes

$$\tilde{\mathcal{L}}(f) = \mathbb{E}_{Y\tilde{Z}}\left[\left(Y - \mathbb{E}_{\tilde{X}|\tilde{Z}}\left[f(\tilde{X})\right]\right)^2\right] + \Omega(f),$$

which is equivalent to the original loss $\mathcal{L}$. This approach is adopted in KIV (Singh et al., 2019), and we used it here for DeepGMM method (Bennett et al., 2019) in the demand design experiment. However, this ignores the fact that we only have to consider the conditional expectation of $X$ given $\tilde{Z} = (Z, O)$. Hence, we introduce another approach which is to model $f(X, O) = \boldsymbol{u}^\top(\boldsymbol{\psi}(X) \otimes \boldsymbol{\xi}(O))$, where $\boldsymbol{\psi}(X)$ and $\boldsymbol{\xi}(O)$ are feature maps and $\otimes$ denotes the tensor product defined as $\boldsymbol{a} \otimes \boldsymbol{b} = \text{vec}(\boldsymbol{a}\boldsymbol{b}^\top)$. It follows that $\mathbb{E}_{X|Z,O}\left[f(X, O)\right] = \boldsymbol{u}^\top(\mathbb{E}_{X|Z,O}\left[\boldsymbol{\psi}(X)\right] \otimes \boldsymbol{\xi}(O))$, which yields the following two-stage regression procedure.

In stage 1, we learn the matrix $\hat{\boldsymbol{V}}$ that

$$\hat{\boldsymbol{V}} = \underset{\boldsymbol{V} \in \mathbb{R}^{d_1 \times d_2}}{\arg\min}\, \mathcal{L}_1(\boldsymbol{V}) \quad \mathcal{L}_1(\boldsymbol{V}) = \mathbb{E}_{X,Z,O}\left[\|\boldsymbol{\psi}(X) - \boldsymbol{V}\boldsymbol{\phi}(Z, O)\|^2\right] + \lambda_1\|\boldsymbol{V}\|^2,$$

which estimates the conditional expectation $\mathbb{E}_{X|Z,O}\left[\boldsymbol{\psi}(X)\right]$. Then, in stage 2, we learn $\hat{\boldsymbol{u}}$ using

$$\hat{\boldsymbol{u}} = \underset{\boldsymbol{u} \in \mathbb{R}^{d_1}}{\arg\min}\, \mathcal{L}_2(w) \quad \mathcal{L}_2(w) = \mathbb{E}_{Y,Z,O}\left[\|Y - \boldsymbol{u}^\top(\hat{\boldsymbol{V}}\boldsymbol{\phi}(Z, O) \otimes \boldsymbol{\xi}(O))\|^2\right] + \lambda_2\|\boldsymbol{u}\|^2.$$

Again, both stages can be formulated as ridge regressions, and thus enjoy closed-form solutions. We can further extend this to learn deep feature maps. Let $\boldsymbol{\phi}_{\theta_Z}(Z,O), \boldsymbol{\psi}_{\theta_X}(X), \boldsymbol{\xi}_{\theta_O}(O)$ be the feature maps parameterized by $\theta_Z, \theta_X, \theta_O$, respectively. Using notation similar to Section 3, the corresponding DFIV algorithm with observable confounders is shown in Algorithm 2. Note that in this algorithm, steps 3, 5, 6 are run until convergence, unlike for Algorithm 1.

---

**Algorithm 2** Deep Feature Instrumental Variable with Observable Confounder

---

**Input:** Stage 1 data $(x_i, z_i, o_i)$, Stage 2 data $(\tilde{y}_i, \tilde{z}_i, \tilde{o}_i)$. Initial values $\hat{\theta}_O, \hat{\theta}_X, \hat{\theta}_Z$. Regularizing parameters $(\lambda_1, \lambda_2)$

**Output:** Estimated structural function $\hat{f}_{\text{struct}}(x)$.

1: **while** $\tilde{\mathcal{L}}_2^{(n)}$ has not converged **do**
2:    **while** $\tilde{\mathcal{L}}_1^{(m)}$ has not converged **do**
3:       Update $\hat{\theta}_Z$ by $\hat{\theta}_Z \leftarrow \hat{\theta}_Z - \alpha \nabla_{\theta_Z} \tilde{\mathcal{L}}_1^{(m)}(\hat{\boldsymbol{V}}^{(m)}(\hat{\theta}_O, \hat{\theta}_X, \theta_Z), \theta_Z)|_{\theta_Z = \hat{\theta}_Z}$    \\ *Stage 1*
4:    **end while**
5:    Update $\hat{\theta}_X$ by $\hat{\theta}_X \leftarrow \hat{\theta}_X - \alpha \nabla_{\theta_X} \tilde{\mathcal{L}}_2^{(n)}(\hat{\boldsymbol{u}}^{(n)}(\hat{\theta}_O, \theta_X, \hat{\theta}_Z), \hat{\theta}_O, \theta_X)|_{\theta_X = \hat{\theta}_X}$    \\ *Stage 2*
6:    Update $\hat{\theta}_O$ by $\hat{\theta}_O \leftarrow \hat{\theta}_O - \alpha \nabla_{\theta_O} \tilde{\mathcal{L}}_2^{(n)}(\hat{\boldsymbol{u}}^{(n)}(\theta_O, \hat{\theta}_X, \hat{\theta}_Z), \theta_O, \hat{\theta}_X)|_{\theta_O = \hat{\theta}_O}$    \\ *Stage 2*
7: **end while**
8: Compute $\hat{\boldsymbol{u}}^{(n)}$ from (9)
9: **return** $\hat{f}_{\text{struct}}(x) = (\hat{\boldsymbol{u}}^{(n)})^\top \boldsymbol{\psi}_{\hat{\theta}_X}(x)$

---

## D   APPLICATION OF 2SLS AND DFIV TO OFF-POLICY POLICY EVALUATION

Here we first show that the optimal solution of (2) in the OPE problem is equivalent to that of mean squared Bellman error, and then describe how we apply 2SLS algorithm.

We interpret the Bellman equation in (10) as IV regression problem, where $X = (s, a, s', a'), Z = (s, a)$ and $Y = r$ and

$$f_{\text{struct}}(s, a, s', a') = Q^\pi(s, a) - \gamma Q^\pi(s', a').$$

Let $\bar{R}(s, a)$ be the conditional expectation of reward given $s, a$ defined as

$$\bar{R}(s, a) = \mathbb{E}_{r|s,a}[r] = \int r P(s'|s, a) R(r|s, a, s') \mathrm{d}s' \mathrm{d}r.$$

Then, we can prove that the solutions of (11) and MSBE are equivalent. Indeed, we have

$$\underset{Q^\pi}{\arg\min} \, \mathbb{E}_{s,a,r}\left[\left(r - Q^\pi(s, a) + \gamma \mathbb{E}_{s',a'|a,s}[Q^\pi(s', a')]\right)^2\right] \quad // \ (11)$$

$$= \underset{Q^\pi}{\arg\min} \, \mathbb{E}_{s,a,r}\left[\left(r - \bar{R}(s, a) + \bar{R}(s, a) - Q^\pi(s, a) + \gamma \mathbb{E}_{s',a'|a,s}[Q^\pi(s', a')]\right)^2\right]$$

$$= \underset{Q^\pi}{\arg\min} \, \mathbb{E}_{s,a}\Big[\text{Var}(r|s, a) \quad // \text{ const wrt } Q^\pi$$

$$+ 2\mathbb{E}_{r|s,a}\left[(r - \bar{R}(s, a))\right]\left(\bar{R}(s, a) - Q^\pi(s, a) + \gamma \mathbb{E}_{s',a'|a,s}[Q^\pi(s', a')]\right) \quad // = 0$$

$$+ \left(\bar{R}(s, a) - Q^\pi(s, a) + \gamma \mathbb{E}_{s',a'|a,s}[Q^\pi(s', a')]\right)^2\Big]$$

$$= \underset{Q^\pi}{\arg\min} \, \mathbb{E}_{s,a}\left[\left(\bar{R}(s, a) - Q^\pi(s, a) + \gamma \mathbb{E}_{s',a'|a,s}[Q^\pi(s', a')]\right)^2\right]. \tag{16}$$

In this context, we model $Q^\pi(s, a) = \boldsymbol{u}^\top \boldsymbol{\psi}(s, a)$ so that

$$f_{\text{struct}}(s, a, s', a') = \boldsymbol{u}^\top(\boldsymbol{\psi}(s, a) - \gamma \boldsymbol{\psi}(s', a')). \tag{17}$$

It follows that

$$\mathbb{E}_{X|z}[f(X)] = \boldsymbol{u}^\top(\boldsymbol{\psi}(s, a) - \gamma \mathbb{E}_{s' \sim P(\cdot|s,a), a' \sim \pi(\cdot|s')}[\boldsymbol{\psi}(s', a')]).$$

We will model $\mathbb{E}_{s' \sim P(\cdot|s,a), a' \sim \pi(\cdot|s')}\left[\boldsymbol{\psi}(s', a')\right] = \boldsymbol{V}\boldsymbol{\phi}(s, a)$. In this case, given stage 1 data $(s_i, a_i, s'_i)$, stage 1 regression becomes

$$\hat{\boldsymbol{V}}^{(m)} = \underset{\boldsymbol{V} \in \mathbb{R}^{d_1 \times d_2}}{\arg\min} \mathcal{L}_1^{(m)}, \quad \mathcal{L}_1^{(m)} = \frac{1}{m} \sum_{i=1}^{m} \|\boldsymbol{\psi}(s_i, a_i) - \boldsymbol{V}\boldsymbol{\phi}(s'_i, a'_i)\|^2 + \lambda_1 \|\boldsymbol{V}\|^2,$$

where we sample $a'_i \sim \pi(\cdot|s'_i)$. However, given the specific form (17) of the structural function, stage 2 is slightly modified and requires minimizing the loss

$$\hat{\boldsymbol{u}}^{(n)} = \underset{\boldsymbol{u} \in \mathbb{R}^{d_1}}{\arg\min} \mathcal{L}_2^{(n)}, \quad \mathcal{L}_2^{(n)} = \frac{1}{n} \sum_{i=1}^{n} (\tilde{r}_i - \boldsymbol{u}^\top (\boldsymbol{\psi}(\tilde{s}_i, \tilde{a}_i) - \gamma \hat{\boldsymbol{V}}^{(m)} \boldsymbol{\phi}(\tilde{s}_i, \tilde{a}_i)))^2 + \lambda_2 \|\boldsymbol{u}\|^2,$$

given stage 2 data $(\tilde{s}_i, \tilde{a}_i, \tilde{r}_i)$. We can further learn deep feature maps by parameterized feature maps $\boldsymbol{\psi}$ and $\boldsymbol{\phi}$ as described in Section 3 to obtain the DFIV algorithm for OPE.

# E    EXPERIMENT DETAILS AND ADDITIONAL RESULTS

## E.1    DETAILS OF DEMAND DESIGN EXPERIMENTS

Here, we introduce the details of demand design experiments. We follow the procedure in Singh et al. (2019). The observations are generated from the IV model,

$$Y = f_{\text{struct}}(P, T, S) + \varepsilon, \quad \mathbb{E}\left[\varepsilon|C, T, S\right] = 0,$$

where $Y$ is sales, $P$ is the treatment variable (price) instrumented by supply cost-shifter $C$. $T, S$ are the observable confounder, interpretable as time of year and customer sentiment. The true structural function is

$$f_{\text{struct}}(P, T, S) = 100 + (10 + P)Sh(T) - 2P,$$
$$h(t) = 2\left(\frac{(t-5)^4}{600} + \exp(-4(t-5)^2) + \frac{t}{10} - 2\right).$$

Data is sampled as

$$\begin{aligned}
S &\sim \text{Unif}\{1, \ldots, 7\} \\
T &\sim \text{Unif}[0, 10] \\
C &\sim \mathcal{N}(0, 1) \\
V &\sim \mathcal{N}(0, 1) \\
\varepsilon &\sim \mathcal{N}(\rho V, 1 - \rho^2) \\
P &= 25 + (C + 3)h(T) + V
\end{aligned}$$

From observations of $(Y, P, T, S, C)$, we estimate $\hat{f}_{\text{struct}}$ by several methods. For each estimated $\hat{f}_{\text{struct}}$, we measure out-of-sample error as the mean square error of $\hat{f}$ versus true $f_{\text{struct}}$ applied to 2800 values of $(p, t, s)$. Specifically, we consider 20 evenly spaced values of $p \in [10, 25]$, 20 evenly spaced values of $t \in [0, 10]$, and all 7 values $s \in \{1, \ldots, 7\}$.

## E.2    EFFECT ESTIMATION IN DEMAND DESIGN EXPERIMENTS

In this section, we report the result of causal effect estimation based on demand design experiments. Specifically, we consider two effects: one is *average treatment effect (ATE)* and another is *conditional average treatment effect (CATE)*.

**ATE Estimation**    ATE is a central target quantity in causal inference defined as

$$\text{ATE} = \mathbb{E}\left[Y|\text{do}(X = 1)\right] - \mathbb{E}\left[Y|\text{do}(X = 0)\right]$$

for binary treatment $X \in \{0, 1\}$. ATE thus requires estimating the counterfactual mean outcomes $\mathbb{E}\left[Y|\text{do}(X = 1)\right]$, $\mathbb{E}\left[Y|\text{do}(X = 0)\right]$. Here, we generalize this to continuous treatment

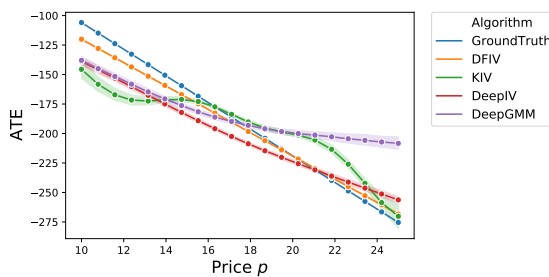

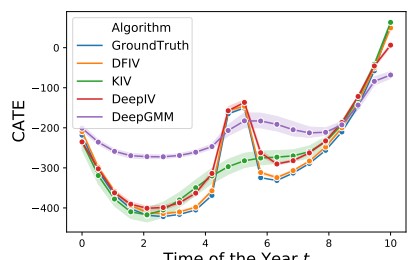

Figure 8: Estimation of ATE.

Figure 9: Estimation of CATE conditioned on $T$ given $P = 25$.

and consider estimating $\mathbb{E}[Y|\text{do}(X = x)]$. This is also known as *continuous treatment effect* or *dose-response curve*.

In the demand design problem, the ground truth is given by

$$\mathbb{E}[Y|\text{do}(P = p)] = \mathbb{E}_{T,S}[f_{\text{struct}}(p, T, S)]$$
$$= (G - 2)p + 100 + 10G,$$

where $G = \mathbb{E}_S[S]\mathbb{E}_T[h(T)]$. The important point here is that this quantity must be monotonically decreasing with respect to price $P$, since we should observe drop of demand as we increase the ticket price.

We estimate ATE by averaging the estimated structural function $\hat{f}_{\text{struct}}$ over $S$ and $T$. The estimated ATE given by DeepGMM, KIV, DeepIV, DFIV are shown in Figure 8. From Figure 8, we can see that KIV and DeepGMM fail to recover the monotonic structure in ATE. DeepIV and DFIV are able to capture this structure, but DFIV estimation is consistently better than DeepIV.

**CATE Estimation**   Although ATE captures the treatment effect for the entire population, treatment effects may be heterogeneous for different sub-populations, which we might be interested in. In such a case, we can consider CATE defined as

$$\text{CATE}(\tilde{o}) = \mathbb{E}\left[Y \mid \text{do}(X = 1), \tilde{O} = \tilde{o}\right] - \mathbb{E}\left[Y \mid \text{do}(X = 0), \tilde{O} = \tilde{o}\right]$$

for binary treatment $X \in \{0, 1\}$. Here, $\tilde{O}$ is a sub-vector of observable confounder $O$. Again, we generalize this idea and consider $\mathbb{E}\left[Y \mid \text{do}(X = x), \tilde{O} = \tilde{o}\right]$, which allows treatment $X$ to be continuous. This quantity is also known as the *heterogeneous treatment effect*.

In demand design experiment, we can consider the CATE conditioned on the time of the year $T$. The true CATE is defined as

$$\mathbb{E}[Y \mid \text{do}(P = p), T = t] = \mathbb{E}_S[f_{\text{struct}}(p, t, S)]$$
$$= 100 + (10 + p)\mathbb{E}[S]h(t) - 2p,$$

which can be obtained by averaging the estimated structural function $\hat{f}_{\text{struct}}$ over $S$. Figure 9 shows the prediction of CATE with respect to $T$ where treatment $P$ is fixed to $P = 25$. Again, we can observe that KIV and DeepGMM fail to capture the shape of the ground truth curve. DeepIV performs significantly better but is less accurate than DFIV.

### E.3   DATA GENERATION PROCESS IN DSPRITES EXPERIMENTS

Here, we describe the data generation process for the dSprites dataset experiment. This is an image dataset described via five latent parameters (`shape, scale, rotation, posX` and `posY`). The images are $64 \times 64 = 4096$-dimensional. In this experiment, we fixed `shape` parameter to `heart`, i.e. we only used the heart-shaped images. The other latent parameters take values of `scale` $\in [0.5, 1]$, `rotation` $\in [0, 2\pi]$, `posX` $\in [0, 1]$, `posY` $\in [0, 1]$.

From this dataset, we generate the treatment variable $X$ and outcome $Y$ as follows.

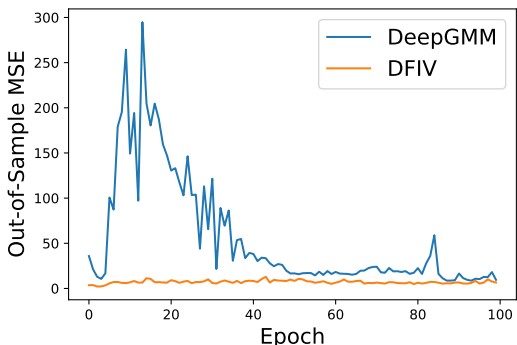

Figure 10: Out-of-Sample MSE in dSprite experiment during the training

1. Uniformly samples latent parameters (`scale`, `rotation`, `posX`, `posY`).

2. Generate treatment variable $X$ as

$$X = \texttt{Fig}(\texttt{scale}, \texttt{rotation}, \texttt{posX}, \texttt{posY}) + \boldsymbol{\eta}.$$

3. Generate outcome variable $Y$ as

$$Y = \frac{\|AX\|_2^2 - 5000}{1000} + 32(\texttt{posY} - 0.5) + \varepsilon.$$

Here, function `Fig` returns the corresponding image to the latent parameters, and $\boldsymbol{\eta}, \varepsilon$ are noise variables generated from $\boldsymbol{\eta} \sim \mathcal{N}(0.0, 0.1I)$ and $\varepsilon \sim \mathcal{N}(0.0, 0.5)$. Each element of the matrix $A \in \mathbb{R}^{10 \times 4096}$ is generated from $\mathrm{Unif}(0.0, 1.0)$ and fixed throughout the experiment. From the data generation process, we can see that $X$ and $Y$ are confounded by `posY`. We use the instrumental variable $Z = (\texttt{scale}, \texttt{rotation}, \texttt{posX}) \in \mathbb{R}^3$, and figures with random noise as treatment variable $X$. The variable `posY` is not revealed to the model, and there is no observable confounder. The structural function for this setting is

$$f_{\mathrm{struct}}(X) = \frac{\|AX\|_2^2 - 5000}{1000}.$$

We use 588 test points for measuring out-of-sample error, which is generated from the grid points of latent variables. The grids consist of 7 evenly spaced values for `posX`, `posY`, 3 evenly spaced values for `scale`, and 4 evenly spaced values for `orientation`.

### E.4  INSTABILITY OF DEEPGMM IN DSPRITES EXPERIMENTS

In the dSprite experiments, we observed that the training procedure of DeepGMM can be unstable. Figure 10 displays the MSE for the models learned in the first 100 epochs. Here, we can see that DeepGMM performs poorly in the early stage of the learning. Furthermore, even after it appears to have converged, we observe a sudden increase of MSE around 80th epoch, which makes difficult to determine when to stop. We conjecture that this is due to the instability of the smooth zero-sum game solved by DeepGMM. By contrast, DFIV converges quickly and performs consistently better than DeepGMM on this task.

### E.5  MNIST EXPERIMENTS

Here, we report the result of MNIST experiments proposed by Bennett et al. (2019), who consider the following datasets:

$$Z \sim \mathrm{Unif}([-3, 3]^2)$$
$$e \sim \mathcal{N}(0, 1), \quad \gamma, \delta \sim \mathcal{N}(0, 0.1)$$
$$X = Z_1 + e + \gamma$$
$$Y = |X| + e + \delta$$

| | MNIST$_x$ | MNIST$_z$ | MNIST$_{xz}$ |
|---|---|---|---|
| DeepGMM | .15 ± .02 | .07 ± .02 | .14 ± .02 |
| DFIV | .18 ± .01 | .07 ± .001 | .10 ± .003 |

Table 1: Out-of-Sample MSE in MNIST experiment of Bennett et al. (2019)

| | Catch | Mountain Car | Cartpole |
|---|---|---|---|
| $D$ | 50 | 3 | 6 |
| $A$ | 3 | 3 | 3 |

Table 2: Dimensions of BSuite tasks

Here, the structural function we aim to learn is $f_{\text{struct}}(X) = |X|$. Additionally, we map $Z$, $X$, or both $X$ and $Z$ to MNIST images to see whether the model can handle the high-dimensional treatment and instrumental variables. Let the output of original IV problem above to be $X_{\text{low}}$, $Z_{\text{low}}$ and $\pi(x) = \text{round}(\min(\max(1.5x + 5, 0), 9))$ be a transformation function that maps inputs to an integer between 0 and 9, and let RandomImage($d$) be a function that selects a random MNIST image from the digit class $d$. The images are $28 \times 28$ = 784-dimensional. We consider the three following scenarios.

- **MNIST$_x$**: $X \leftarrow X_{\text{low}}, Z \leftarrow \text{RandomImage}(\pi(Z_{\text{low}}))$
- **MNIST$_z$**: $X \leftarrow \text{RandomImage}(\pi(X_{\text{low}})), Z \leftarrow Z_{\text{low}}$
- **MNIST$_{xz}$**: $X \leftarrow \text{RandomImage}(\pi(X_{\text{low}})), Z \leftarrow \text{RandomImage}(\pi(Z_{\text{low}}))$

We refer the reader to Bennett et al. (2019) for a detailed description. We applied DFIV using the same architecture as DeepGMM. In Table 1, we present the mean and standard error of out-of-sample MSE for DFIV and report the results obtained for DeepGMM in Bennett et al. (2019). From Table 1, we can see that DFIV performs essentially like DeepGMM.

### E.6 OPE EXPERIMENT DETAILS

#### E.6.1 BSUITE TASKS

We provide a brief description below of the three behavior suite (BSuite) reinforcement learning tasks in Osband et al. (2019):

1. Catch: A 10x5 Tetris-grid with single block falling per column. The agent can move left/right in the bottom row to 'catch' the block. Illustrated in Figure 11a.

2. Mountain Car: The agent drives an underpowered car up a hill (Moore, 1990). Illustrated in Figure 11b.

3. Cartpole: The agent can move a cart left/right on a plane to keep a balanced pole upright (Barto et al., 1983). Illustrated in Figure 11c.

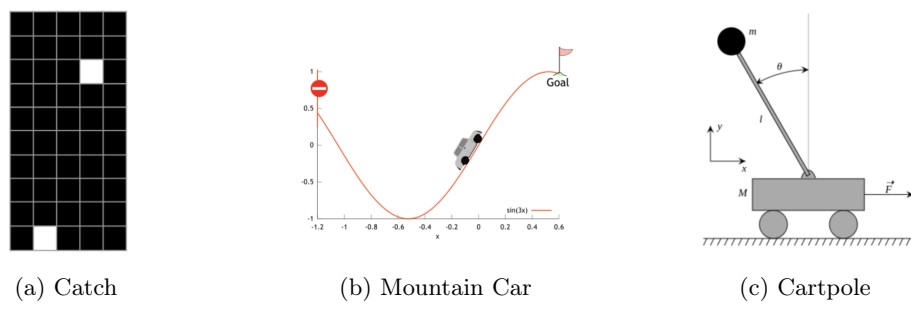

(a) Catch        (b) Mountain Car        (c) Cartpole

Figure 11: Three BSuite tasks. Figures are from Osband et al. (2019).

All tasks have a real-valued state space $\mathcal{S} \subseteq \mathcal{R}^D$ and a discrete action space $\mathcal{A} = \{0, 1, 2, \ldots, A - 1\}$. The state dimension and number of actions are provided in Table 2.

### E.6.2 Fitted Q Evaluation

Fitted Q Evaluation (FQE) (Le et al., 2019) is a simple variant of the Fitted Q Iteration (FQI) algorithm (Ernst et al., 2005). Instead of learning the Q function of an optimal policy as FQI, FQE estimates the Q function of a fixed policy $\pi$. It is an iterative algorithm with randomly initialized the Q function parameters $\theta_0$. At iteration $k \geq 1$ with the current estimate of Q function $Q^\pi(s, a | \theta_{k-1})$, it updates parameters $\theta_k$ by solving the following regression problem using a least squares approach:

$$
\begin{aligned}
r + \gamma Q^\pi(s', a' | \theta_{k-1}) = & Q^\pi(s, a | \theta_k) \\
& + \gamma \left( Q^\pi(s', a' | \theta_{k-1}) - \mathbb{E}_{s' \sim P(\cdot | s, a), a' \sim \pi(\cdot | s')} \left[ Q^\pi(s', a' | \theta_{k-1}) \right] \right) \\
& + r - \mathbb{E}_{r \sim R(\cdot | s, a, s')} \left[ r \right],
\end{aligned}
\tag{18}
$$

where the regression function to estimate is $Q^\pi(s, a | \theta_k)$, the observed outcome is the term on the LHS of (18) and the residual is the sum of terms in the second and third lines of (18). Comparing the equation above with (10), FQE reformulates the regression problem and moves the confounded part in the treatment of (10), that is $\gamma Q^\pi(s', a')$, to the outcome. The regression problem at each iteration is therefore not confounded any more. If FQE converges, it finds a solution of the following equation

$$
Q^\pi(s, a | \theta) = P(\mathbb{E}_{r | s, a} \left[ r \right] + \gamma \mathbb{E}_{s', a' | s, a} \left[ Q^\pi(s', a' | \theta) \right]),
\tag{19}
$$

where $P(\cdot)$ is the $L_2$ projection operator that maps a function of $(s, a)$ onto the (parameterized) Q function space.

## F Failure of Joint Optimization

One might want to jointly minimize the loss $\tilde{\mathcal{L}}_2^{(n)}$, which is the empirical approximation of $\mathcal{L}$. However, this fails to learn the true structural function $f_{\text{struct}}$, as shown in this appendix.

Figure 12 shows the learning curve obtained when we jointly minimize $\theta_X$ and $\theta_Z$ with respect to $\tilde{\mathcal{L}}_2^{(n)}$, where $\mathcal{L}_{\text{test}}$ is the empirical approximation of

$$
L_{\text{test}} = \mathbb{E}_X \left[ \| f_{\text{struct}}(X) - (\hat{\boldsymbol{u}}^{(n)})^\top \boldsymbol{\psi}_{\theta_X}(X) \|^2 \right].
\tag{20}
$$

We observe in Figure 12 that the decrease of stage 2 loss does not improve the performance of the learned structural function. This is because the model focuses on learning the relationship between instrument $Z$ and outcome $Y$, while ignoring treatment $X$. We can see this from the fact that stage 1 loss becomes large and unstable. This is against the goal of IV regression, which is to learn a causal relationship between $X$ and $Y$.

On the other hand, Figure 13 shows the learning curve obtained using DFIV. Now, we can see that stage 2 loss matches $\mathcal{L}_{\text{test}}$. Also, we can confirm that stage 1 loss stays small and stable.

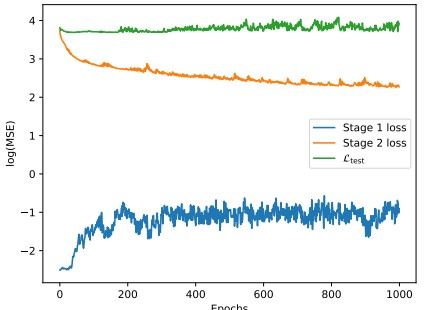

Figure 12: Learning curve of joint minimization

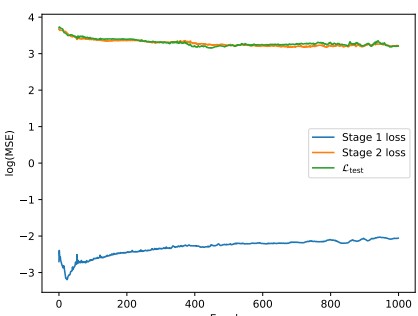

Figure 13: Learning curve of DFIV

Table 3: Network structures of DeepIV for demand design dataset. For the input layer, we provide the input variable. For the fully-connected layers (FC), we provide the input and output dimensions. For mixture Gaussian output, we report the number of components. Dropout rate is given in the main text.

| Instrument Net | | Treatment Net | |
|---|---|---|---|
| Layer | Configuration | Layer | Configuration |
| 1 | Input$(C, T, S)$ | 1 | Input$(P, T, S)$ |
| 2 | FC(3, 128), ReLU | 2 | FC(3, 128), ReLU |
| 3 | Dropout | 3 | Dropout |
| 4 | FC(128, 64), ReLU | 4 | FC(128, 64), ReLU |
| 5 | Dropout | 5 | Dropout |
| 6 | FC(64, 32), ReLU | 6 | FC(64, 32), ReLU |
| 7 | Dropout | 7 | Dropout |
| 8 | MixtureGaussian(10) | 8 | FC(32, 1) |

## G   NETWORK STRUCTURES AND HYPER-PARAMETERS

Here, we describe the network architecture and hyper-parameters of all experiments. Unless otherwise specified, all neural network-based algorithms are optimized using Adam with learning rate = 0.001, $\beta_1 = 0.9$, $\beta_2 = 0.999$ and $\varepsilon = 10^{-8}$.

**Demand Design**   For DeepIV, we used the original structure proposed in Hartford et al. (2017), which is described in Table 3. We follow the default dropout rate in Hartford et al. (2017), which depends on the data size. For DFIV, we used the structure described in Table 4. The regularizer $\lambda_1, \lambda_2$ are both set to 0.1 as a result of the tuning procedure described in Appendix A. For KIV, we used the Gaussian kernel where the bandwidth is determined by the median trick described by Singh et al. (2019). We used random Fourier feature trick (Rahimi and Recht, 2008) with 100-dimensions. For DeepGMM, we used the same structure as DeepIV but no dropout is applied and the last layer of the Instrument Net is changed to a fully-connected layer which maps 32 dimensions to 1 dimension.

**Demand Design with MNIST**   The feature extractor for MNIST image data is given in Table 5, which is used for both stage 1 and 2. For DeepIV, we used the original structure proposed in Hartford et al. (2017), which is described in Table 6. We follow the default dropout rate in Hartford et al. (2017), which depends on the data size. For DFIV, we used the structure described in Table 7. The regularizer $\lambda_1, \lambda_2$ are both set to 0.1 as a result of the tuning procedure described in Appendix A. For KIV, we used the Gaussian kernel where the bandwidth is determined by the median trick and also used random Fourier features with 100-dimensions. For DeepGMM, we used the same structure as DeepIV but no dropout

Table 4: Network structures of DFIV for demand design datasets. For the input layer, we provide the input variable. For the fully-connected layers (FC), we provide the input and output dimensions.

**Instrument Feature $\phi_{\theta_Z}$**

| Layer | Configuration |
|---|---|
| 1 | Input($C, T, S$) |
| 2 | FC(3, 128), ReLU |
| 3 | FC(128, 64), ReLU |
| 4 | FC(64, 32), ReLU |

**Treatment Feature $\psi_{\theta_X}$**

| Layer | Configuration |
|---|---|
| 1 | Input($P$) |
| 2 | FC(1, 16), ReLU |
| 3 | FC(16, 1) |

**Observable Feature $\xi_{\theta_O}$**

| Layer | Configuration |
|---|---|
| 1 | Input($T, S$) |
| 2 | FC(2, 128), ReLU |
| 4 | FC(128, 64), ReLU |
| 6 | FC(64, 32), BN, ReLU |

Table 5: Network structures of feature extractor used in demand design experiment with MNIST. For each convolution layer, we list the input dimension, output dimension, kernel size, stride, and padding. For the input layer, we provide the input variable. For the fully-connected layers (FC), we provide the input and output dimensions. For max-pool, we list the size of the kernel. Dropout rate here is set to 0.1. SN denotes Spectral Normalization (Miyato et al., 2018).

**ImageFeature**

| Layer | Configuration |
|---|---|
| 1 | Input($S$) |
| 2 | Conv2D (1, 64, 3, 1, 1), ReLU, SN |
| 3 | Conv2D (64, 64, 3, 1, 1), ReLU, SN |
| 4 | MaxPool(2, 2) |
| 5 | Dropout |
| 6 | FC(9216, 64), ReLU |
| 7 | Dropout |
| 8 | FC(64, 32), ReLU |

is applied and the last layer of instrument net is changed to a fully connected layer which maps 32 dimensions to 1 dimension.

**dSprite experiment**  For DeepGMM, we used the structure described in Table 8. For DFIV, we used the structure described in Table 9. The regularizer $\lambda_1, \lambda_2$ are both set to 0.01 as a result of the tuning procedure described in Appendix A. For KIV, we used the Gaussian kernel where the bandwidth is determined by the median trick. We used random Fourier feature trick (Rahimi and Recht, 2008) with 100 dimensions.

**OPE experiment**  For DeepIV, we used the structure described in Table 12. For DFIV, we used the structure described in Table 10. The regularizer $\lambda_1, \lambda_2$ are both set to $10^{-5}$ as a result of the tuning procedure described in Appendix A. For KIV, we used the Gaussian kernel where the bandwidth is determined by the median trick. We used random Fourier features (Rahimi and Recht, 2008) with 100 dimensions. For DeepGMM, we use the structure described in Table 11.

Table 6: Network structures of DeepIV in demand design with MNIST data. For the input layer, we provide the input variable. For the fully-connected layers (FC), we provide the input and output dimensions. For mixure Gaussian output, we report the number of components. ImageFeature denotes the module given in Table 5. Dropout rate is described in the main text.

| **Instrument Net** | |
|:---:|:---:|
| Layer | Configuration |
| 1 | $\mathrm{Input}(C, T, \mathtt{ImageFeature}(S))$ |
| 2 | FC(66, 32), ReLU |
| 3 | Dropout |
| 4 | MixtureGaussian(10) |

| **Treatment Net** | |
|:---:|:---:|
| Layer | Configuration |
| 1 | $\mathrm{Input}(P, T, \mathtt{ImageFeature}(S))$ |
| 2 | FC(66, 32), ReLU |
| 3 | Dropout |
| 4 | FC(32, 1), ReLU |

Table 7: Network structures of DFIV in demand design with MNIST. For the input layer, we provide the input variable. For the fully-connected layers (FC), we provide the input and output dimensions. ImageFeature denotes the module given in Table 5.

| **Instrumental Feature $\phi_{\theta_Z}$** | |
|:---:|:---:|
| Layer | Configuration |
| 1 | $\mathrm{Input}(C, T, \mathtt{ImageFeature}(S))$ |
| 2 | FC(66, 32), BN, ReLU |

| **Treatment Feature $\psi_{\theta_X}$** | |
|:---:|:---:|
| Layer | Configuration |
| 1 | $\mathrm{Input}(P)$ |
| 2 | FC(1, 16), ReLU |
| 3 | FC(16, 1) |

| **Obsevable Feature Net $\xi_{\theta_O}$** | |
|:---:|:---:|
| Layer | Configuration |
| 1 | $\mathrm{Input}(T, \mathtt{ImageFeature}(S))$ |
| 2 | FC(65, 32), BN, ReLU |

Table 8: Network structures of DeepGMM in dSprite experiment. For the input layer, we provide the input variable. For the fully-connected layers (FC), we provide the input and output dimensions. SN denotes Spectral Normalization (Miyato et al., 2018).

| **Instrument Net** | |
|:---:|:---:|
| Layer | Configuration |
| 1 | $\mathrm{Input}(Z)$ |
| 2 | FC(3, 256), SN, ReLU |
| 3 | FC(256, 128), SN, ReLU, BN |
| 4 | FC(128, 128), SN, ReLU, BN |
| 5 | FC(128, 32), SN, BN, ReLU |
| 6 | FC(32, 1) |

| **Treatment Net** | |
|:---:|:---:|
| Layer | Configuration |
| 1 | $\mathrm{Input}(X)$ |
| 2 | FC(4096, 1024), SN, ReLU |
| 3 | FC(1024, 512), SN, ReLU, BN |
| 4 | FC(512, 128), SN, ReLU |
| 5 | FC(128, 32), SN, BN, Tanh |
| 6 | FC(32,1) |

Table 9: Network structures of DFIV in dSprite experiment. For the input layer, we provide the input variable. For the fully-connected layers (FC), we provide the input and output dimensions. SN denotes Spectral Normalization (Miyato et al., 2018).

| **Instrument Feature $\phi_{\theta_Z}$** | |
|:---:|:---:|
| Layer | Configuration |
| 1 | $\mathrm{Input}(Z)$ |
| 2 | FC(3, 256), SN, ReLU |
| 3 | FC(256, 128), SN, ReLU, BN |
| 4 | FC(128, 128), SN, ReLU, BN |
| 5 | FC(128, 32), SN, BN, ReLU |

| **Treatment Feature $\psi_{\theta_X}$** | |
|:---:|:---:|
| Layer | Configuration |
| 1 | $\mathrm{Input}(X)$ |
| 2 | FC(4096, 1024), SN, ReLU |
| 3 | FC(1024, 512), SN, ReLU, BN |
| 4 | FC(512, 128), SN, ReLU |
| 5 | FC(128, 32), SN, BN, Tanh |

Table 10: Network structures of DFIV in OPE experiment. For the input layer, we provide the input variable. For the fully-connected layers (FC), we provide the input and output dimensions.

**Instrument Feature $\phi_{\theta_z}$**

| Layer | Configuration |
|-------|---------------|
| 1 | Input($s$, one-hot($a$)) |
| 2 | FC(*, 150), ReLU |
| 3 | FC(150, 100), ReLU |
| 4 | FC(100, 50), ReLU |

**Treatment Feature $\psi_{\theta_x}$**

| Layer | Configuration |
|-------|---------------|
| 1 | Input($s$, one-hot($a$)) |
| 2 | FC(*, 50), ReLU |
| 3 | FC(50, 50), ReLU |

Table 11: Network structures of DeepGMM in OPE experiment. For the input layer, we provide the input variable. For the fully-connected layers (FC), we provide the input and output dimensions.

**Instrument Net**

| Layer | Configuration |
|-------|---------------|
| 1 | Input($s$, one-hot($a$)) |
| 2 | FC(*, 150), ReLU |
| 3 | FC(150, 100), ReLU |
| 4 | FC(100, 50), ReLU |
| 5 | FC(50, 1) |

**Treatment Net**

| Layer | Configuration |
|-------|---------------|
| 1 | Input($s$, one-hot($a$)) |
| 2 | FC(*, 50), ReLU |
| 3 | FC(50, 50), ReLU |
| 4 | FC(50, 1), ReLU |

Table 12: Network structures of DeepIV in OPE experiment. For the input layer, we provide the input variable. For the fully-connected layers (FC), we provide the input and output dimensions. The MixutureGaussian layer maps the input linearly to required parameter dimensions for a mixture of Gaussian distribution with diagonal covariance matrices. The Bernoulli layer maps the input linearly to a single dimension to represent the logit of a Bernoulli distribution to predict if the next state is a terminating state.

**Instrument Net**

| Layer | Configuration |
|-------|---------------|
| 1 | Input($s$, one-hot($a$)) |
| 2 | FC(*, 50), ReLU |
| 3 | FC(50, 50), ReLU |
| 4 | MixtureGaussian(3) and Bernoulli |

**Treatment Net**

| Layer | Configuration |
|-------|---------------|
| 1 | Input($s$, one-hot($a$)) |
| 2 | FC(*, 50), ReLU |
| 3 | FC(50, 50), ReLU |
| 4 | FC(50, 1), ReLU |

