# OpenReview forum: "Learning Deep Features in Instrumental Variable Regression"
_ICLR.cc/2021/Conference — ICLR 2021 Poster_

### Official Review · AnonReviewer3 · 2020-10-28
**interesting paper; experiments could be improved a bit.**

**Rating:** 7
**Confidence:** 4

**Review:**

I liked reading this paper. It is reasonably well-written and tackles an important problem in an interesting way. I list the issues I had with the paper below. I believe one advantage of DFIV is that it does not rely on some relaxed optimization problem like DeepIV. The main idea in the paper is learning a set of basis functions such that the structural function is a linear combination on them; the learning itself relies on predicting the basic functions from the IV which ensures that confounding information is projected out.

First, while I think the method is well motivated, I am confused by the discussion around 'forbidden regression'. I believe the statement made in the paper about 'high variance' may be misleading. 'Forbidden regression' is a statement about the mis-specification of the conditional density model which may lead to an identification failure. Can the authors point out the exact part of Angrist where high variance is called the 'forbidden regression' problem?

I think the proposed method does not face the forbidden regression problem because of the linear relationship between the IV and the outcome implied by the proposed model.

Second, I believe it would be in support of the method to expand the experiments to include the high-dimensional IV and treatment experiments from DeepGMM.

I would like the authors to clarify the following:

1. In 4.2, if the image is given as treatment to the model, isn't the confounder posY be specified as a function of the treatment?
2. In the OPE experiment, the DFIV (and DeepGMM to a lesser extent) does not  exhibit monotonic behavior with noise increase. Could the authors explain if this is randomness? if it is not, could the authors provides results over a larger number of seeds?

Finally, could the authors expand on any convergence issues during training due to the coupled optimization and how they fixed them?

---

> ### Author Response · Authors · 2020-11-19
> **[Manuscript Updated] Thank you for the constructive feedback.**
>
> Thank you for the constructive feedback. We updated the manuscript according to your feedback.
>
> 1. **"Forbidden regression" terminology**:   Thank you for pointing this out. We agree that our use of the term "forbidden regression'' has been imprecise (in fact, there are now multiple definitions of ``forbidden regression'' in the literature). We have removed it entirely from our document in the revised version. Instead, we now make the specific point that learning high-dimensional conditional density models is difficult.
>
> 2. **High-dimensional IV and treatment experiments from DeepGMM**:  Thank you for the suggestion. We have added these experiments to Appendix D.5.
>
> 3. **posY can be specified as a function of the treatment**:   Yes, we can indeed reconstruct confounder $\mathtt{posY}$ from image $X$ (although not perfectly as we have an additive noise $\eta$). However, this does not violate any assumption.
>
> 4. **OPE experiment**:   An OPE problem is specified by one fixed dataset and one target policy for evaluation. For every environment and noise level combination, we generate one OPE problem. Then we run every algorithm for multiple runs with different random seeds. Therefore, we do not expect to observe monotonic behavior with the noise increase due to the randomness in the dataset and policy generation. We compare all methods within the same setting instead. The combination of 3 environments and 6 noise levels provides a diverse set of settings for empirical evaluation. Regarding the coupled optimization, we update stage 1 feature more frequently than stage 2 feature so that the conditional expectation is well-approximated during the training.

---

> > ### Comment · AnonReviewer3 · 2020-11-24
> > **updated score after author response**
> >
> > I'm updating score assuming that the authors will add a complete discussion about identification as per the other reviewers.
> >
> > My confidence in the papers comes from the fact that I believe what the authors construct is an adaptive version of the the basis function based construction in Newey and Powell (https://eml.berkeley.edu/~powell/npiv.pdf).

---

### Official Review · AnonReviewer2 · 2020-10-28
**Nice paper!**

**Rating:** 8
**Confidence:** 5

**Review:**

The main idea in this paper is very nice and is as follows:

Suppose we have the standard IV setup which is
X = m(U) + f(Z)
Y = q(U) + b(X)

Where U is some unobserved confounder.

Typically we are interested in the linearized version of this model. If we regress Y on X, we get a biased estimate of B(X) because U is a common unobserved cause. We can deal with this by first regressing X on Z, then regressing Y on the predicted values of X given Z. It is well known that using any other method than linear regression in this context leads to "forbidden regression" problems. So, typically people just estimate the average effect.

However, with bigger data we might be interested in trying to get more precise estimates of the function b(X) and also f(Z). Typically this is done by hand-crafting a basis (interacting instruments with things, usually). The authors propose using deep networks to learn linear bases and then essentially do 2SLS on top of them (technically this is all trained together, but I am simplifying for the review).

Overall, while I think there are many unanswered questions in this paper, I think it should be accepted for the community to build on it. IV is a tricky area and there is no way to show everything we need to know about a method in one paper.

Below I list several questions I had about the paper which I think the authors could incorporate in various formats (or educate me on how they have already answered them and I missed it):

- Assumptions required
The authors state that the only assumption that is required is the standard exclusion restriction. I feel like that is not necessarily correct. For example, in 2SLS if we assume underlying heterogeneity (which, we must assume in this case otherwise why would we be learning a basis for this heterogeneity?) then if it the heterogeneity is not proxied by observed variables correctly we need some additional assumptions like monotonicity to get an interpretation of the IV estimate as some kind of causal effect (e.g. a LATE). Similarly, even with no covariates if we're trying to estimate the full function b(X), don't we need some assumption about how the IV affects X in the whole support in order to guarantee that we can use the IV to learn b(X)?

- Regularization parameters
As I understand it, and the authors can correct me if I'm wrong, the hyperparameters for the regularizers are set by holding out some (x,z) and some (y, x) and then evaluating standard out of sample predictive loss? This is also what is done in DeepIV. This has been shown to not be an optimal way of choosing hyperparameters in Peysakhovich and Eckles (2018) which talks about "Causal Cross-validation" for adapting split-sample IV ideas from (Angrist & Krueger, 1995; Imbens et al., 1999; Hansen & Kozbur, 2014). The setting in that paper is specifically discrete instruments rather than continuous ones, so that procedure can't exactly be adapted in all the cases here, but it still may be worthwhile to mention as a future direction. Overall, hyperparameter tuning for these problems is really important and even more so when you have these deep feature generators.

- Comparison to control functions / 2 stage residual inclusion
Currently the only comparison made are to other 2sls type methods where endogenous variables are changed to their predicted values. There is another way to estimate IV which is the control function/2 stage residual inclusion approach which seems to lend itself naturally to neural networks/function approximators since all you need to do is include the residual from the first stage into the second stage to get a correct estimate of b(X). It seems like it would be pretty easy to add this as a baseline to at least the high-D demand experiment.

Typo:
The abstract states: "In this case, deep neural nets are trained to define informative nonlinear features on the instruments and treatments." I believe this should say "linear features" in the sense that you are learning the correct basis for 2SLS?

---

> ### Author Response · Authors · 2020-11-19
> **[Manuscript Updated] Thank you for the constructive feedback.**
>
> Thank you for the constructive feedback. We updated the manuscript according to your feedback.
>
> 1. **Assumption required for IV**: You are right. We have now mentioned the conditions provided by Newey and Powell (Econometrica, 2003) to guarantee identifiability of the structural function (this paper is also referenced in DeepIV (Hartford et al. 2017) and DeepGMM (Bennett et al. 2019).
>
> 2. **Regularization parameters**:  Thank you for the references! We have mentioned the idea of adapting some of the methods you mentioned to DFIV in the conclusion of the paper. Certainly, our current hyperparameter learning approach is only a first step, and we are eager to explore the suggested methods to improve it in our future work.
>
> 3. **Comparison to control functions / 2 stage residual inclusion**:  In our understanding, 2 stage residual inclusion (2SRI) methods solves a slightly different problem from our 2SLS methods but please correct us if we are wrong. In 2SRI methods, we fit the structural model $Y = f(X,\varepsilon)$, where $X$ is the treatment and $\varepsilon$ is the unobserved confounder. The idea of 2SRI is to assume $X$ to be $X = g(Z) + \varepsilon$ and replace $\varepsilon$ to the residual of the stage 1 $\hat{\varepsilon} = X - \hat{g}(Z)$. Although 2SRI method can handle nonlinear confounding, we need additional information about $\varepsilon$ or $Z$ when we make a prediction of structural function $f$. On the other hand, 2SLS methods assume that the structural function  depends only on $X$ and the confounder $\varepsilon$ affects the outcome in a additive manner, i.e. $Y = f(X) + \varepsilon$. This assumption enables us to directly compute $f(X)$ without additional information. Considering this difference, we believe 2SRI to be out of the scope of the paper, though learning deep feature in 2SRI would be an interesting direction for future investigation.

---

### Official Review · AnonReviewer1 · 2020-10-29
**Interesting Idea; Conservative Score**

**Rating:** 6
**Confidence:** 2

**Review:**

The authors consider the problem of learning the structural equation governing the relationship between a treatment and outcome in a causal model. They assume an instrumental variable setting and use neural networks to fit non-linear models as part of the conventional 2SLS approach to parameter estimation in IV models.

In general I found the paper to be pretty clearly written and I think it shows promise. That said, there are some points of confusion/clarification discussed below. In light of these, I've opted to conservatively score this paper as a weak reject for the pre-rebuttal/pre-discussion phase and will reevaluate and consider raising my score later in the review process.

Detailed comments (major concerns denoted by **):

In the intro (paragraph 3) the authors describe the characteristics necessary for something to be a valid instrument. I recommend also adding that the instrument Z cannot have an incoming edge from the latent confounder that is also a parent of the outcome.

In the intro the authors state "Although these emthods enjoy desireable theoretical properties, the flexibility of the model is limited, since existing work uses prespecified features". Can the authors clarify what's meant by "prespecified"? It's not obvious how exactly this is a limitation.

In a similar vein, later in the intro, the authors refer to DeepGMM as having an "unstable" learning procedure since it involves solving a smooth zero-sum game. The authors should clarify why this is the case, since many readers, especially in the more traditionalist IV community, may not be familiar with the subtleties of recent competitor methods like DeepGMM.

**How does the assumption of additive noise constrain the problem? What happens if this assumption is dropped? Is the structural function then not identifiable? The authors don't seem to use this assumption later in their argumentation for why their method works and so I'm curious whether either i) the assumption isn't necessary or ii) the assumption is trivializing the problem in some way. In general additive noise is a pretty strong assumption and so this could drastically limit the efficacy of the proposed method.

In line with above comments about clarifying background details about competitor models, can the authors provide more background on Hermite polynomials?

**I'm a bit concerned about the use of regularization in the fitting procedure (Equations 6 and 8). Both of these models are _nuisance_ models. While the authors frame the problem as trying to learn the structural function that governs the X->Y relationship, ultimately what a practitioner cares about is evaluating effects like E[Y(x) - Y(x')]. Traditionally, one has to be very careful about employing regularization in causal modeling since you might regularize an effect to zero in the service of trying to improve prediction performance for the nuisance model. As a toy example, consider a simple linear causal model where Y = Beta X. If the true effect for some element of X, say X_2, is small but nonzero (i.e. epsilon > Beta_2 > 0), then regularizing could make us learn a \hat{Beta} such that \hat{Beta}_2 == 0 which is a biased effect estimate. Perhaps the dynamics are different here because of non-linearities and the regularization does help empirically with the problem the authors consider. But in general I'm not sure that's the case -- the authors should clarify this point in their argumentation and think carefully about this issue in the context of their experiments.

**Some questions about sample efficiency. The authors give some commentary on the computational complexity. This is certainly important, but typically of equal or greater concern is whether the models can actually converge to the true parameter. As far as I know, there are few (if any) results on the sample efficiency of NNs in general and so it's hard (impossible) to say whether or not they are consistent (converge to an unbiased estimator of the true parameter). Is there a sense in which this can be known for the class of NNs the authors consider? What is the rate of convergence of the fitting procedure?

In the airplane experiment, why does time range from 0 to 10 rather than, say 0 to 11 or 1 to 12?

Is there a substantive interpretation for rho in the airplane example? Is there a reasonable latent confounder that this might correspond to that does not act as a causal parent of Z? I realize that it exists as an artifact of the simulation DGP from the Hartford paper but it makes the example somewhat weaker that it might not be something that can be mapped to a fully realistic modeling problem.

"This may be due to the current DeepGMM approach to handling observable confounders, which may not be optimal" -- what is the 'current approach' and why is it suboptimal?

**In the DSprites experiment, what is the outcome Y?

**The authors should strongly consider adding an experiment in which they actually intervene on X. The task of learning f(X) is very close to estimating an effect, but I'm curious to know i) how well their algorithm actually does for effect estimation and ii) whether their model's outperformance relative to competitors is as drastic in that situation. In that case, because they are using a synthetic DGP, it would be possible to know the ground truth effect of the intervention and thus they'd have a principled way to evaluate the performance on the effect estimation task.

To the other reviewers/ACs, I'm not really qualified to evaluate the experiment and claims discussed in section 4.3 regarding off-policy evaluation. To the authors, however, in line with above comments about defining terms from the literature, many readers will not have a clue what things like 'catch and mountain car' are or what the details of the FQE model are and so you should strongly consider clarifying the task and details of the competitor model if space allows.

---

> ### Author Response · Authors · 2020-11-19
> **[Manuscript Updated, 1/2] Thank you for the constructive feedback.**
>
> Thank you for the constructive feedback. We updated the manuscript according to your feedback.
>
> 1.  **Characteristics necessary for something to be a valid instrument**: Thank you for the suggestion. We have added it to the paper.
>
> 2. **Prespecified feature**:  The empirical performance of the methods mentioned therein (Sieve IV, Kernel IV and Dual IV) is strongly dependent on the choice of the features/kernel which are not learned from the data. Our approach can be thought of as an extension of these approaches where the features are also learned from data. For high dimensional $X$ such as images, learned deep features are known to be more effective than pre-specified, fixed kernel features.
>
> 3. **"Unstable" learning procedure in DeepGMM**:  DeepGMM requires solving a ``min-max'' optimization problem similar to those encountered in the Generative Adversarial Networks (GAN) literature. The non-convex-concave objective function makes it especially hard for the gradient descent ascent algorithm to converge, often resulting in diverging, oscillating, or cyclic behavior [1]. The method proposed in the DeepGMM paper to solve the optimization problem actually used an algorithm (OAdam) originally developed for GAN. Practically, we observed that DeepGMM can take a long time to converge. Empirically, it also fairly unstable in the early stage of the training in our dSprite experiments. We have mentioned [1] in the introduction and added some additional experiments in Appendix D.4 illustrating this point.
>
> [1] Wiatrak, M., Albrecht, S.V. and Nystrom, A., 2019. Stabilizing Generative Adversarial Networks: A Survey. arXiv preprint arXiv:1910.00927.
>
> 4. **Additive noise constraint**:  The loss (2) actually depends on the additive noise assumption, since it tries to find $f$ such that $\mathbb{E}[Y|Z] = \mathbb{E}[f(X)|Z]$, which only holds when the confounder is additive. This assumption is common in all IV methods, yet it is interesting to consider more general noise settings.
>     On the one hand, in Bareinboim and Pearl (2012), it is shown that we *cannot* identify the causal effect of $X \to Y$ in our graph if we consider completely general confounding. However, there exist generalizations of the IV setting which permit specific alternative noise forms: for instance, in Section 5.4 of [2], multiplicative noise is considered. The adaptation of DFIV to these alternative noise settings is an interesting topic of future study. We have included this point in the discussion of the revised version.
>
> [2] Marine Carrasco, Jean-Pierre Florens, Eric Renault, Linear Inverse Problems in Structural Econometrics Estimation Based on Spectral Decomposition and Regularization,
> Handbook of Econometrics, Chapter 77, Editor(s): James J. Heckman, Edward E. Leamer, Elsevier, Volume 6, Part B, 2007.
>
> 5. **Hermite polynomials**:  The Hermite polynomials are a fixed basis, which are orthogonal with respect to a Gaussian measure. By contrast, our basis functions are learned, and should be more appropriate to high dimensional data such as images.
>
> 6. **Ridge regularization**:  It is true that regularization induces a bias and might ignore small but nonzero effects.  For a fixed basis, as in [Singh et al., NeurIPS 2019], the regularization is used to trade off bias and variance, and is reduced as sample size increases to ensure consistency - in other words, with sufficient samples, the small effects are recovered. In the case of learned features, we conjecture that a similar effect will apply. As a practical matter, we need this regularization in order to stabilize matrix inversion during the learning process, and this is necessary to obtain satisfactory performance.
>
> 7. **Sample efficiency**:  We certainly agree that in neural network methods, the practice is running somewhat ahead of the theory. For this reason, we do not have available consistency results as yet. This is an important future direction for the field. Nonetheless, a present practical advantage of neural nets is their strong empirical performance on high dimensional and complex data.
>
> 8. **Time range from 0 to 10**: This follows the previously proposed data generation mechanism in Harford et al., which we use for consistency with the prior work.
>
> 9. **interpretation for rho in airplane example**: We are not aware of a substantive interpretation for rho in the airplane example and it is indeed taken from the Hartford paper. In the final RL example, the level of confounding has a clear interpretation as it corresponds to the level of stochasticity of the policy.

---

> > ### Author Response · Authors · 2020-11-19
> > **[Manuscript Updated, 2/2] Thank you for the constructive feedback.**
> >
> > 10. **"This may be due to the current DeepGMM approach to handling observable confounders, which may not be optimal" -- what is the 'current approach' and why is it suboptimal?**: In the DeepGMM paper, the authors suggest adding the observable confounders to both the treatment variables and instrumental variables. This is valid but it ignores the fact that the conditional expectation of the observable confounders given the instrumental variables is known in this case. DFIV is able to exploit this property to solve an easier (lower-dimensional) regression problem (see Appendix B) while DeepGMM still relies on optimizing over test functions $f$ which are functions of both unobserved and observable confounders.
> >
> > 11. **Outcome in dSprite experiment**: Sorry for the confusion. The outcome is defined as $$Y = \frac{\|AX\|_2^2 - 5000}{1000} + 32(\mathtt{posY}-0.5) + \varepsilon,$$  where $A$ is a random matrix fixed throughout the experiment. This was described in the appendix, but we have moved it to the main text in the revised version.
> >
> > 12. **Adding an experiment for estimating effects**: Thank you for the suggestion. We have added the experiment on average treatment effect (ATE) and conditional average treatment effect (CATE) to Appendix D.2.
> >
> > 13. **off-policy evaluation experiments**: Thanks for pointing this out. We have added more details on the RL problem and methodology in the paper and the appendix. We believe that showing that off-policy evaluation can be recast as an IV regression problem is an important contribution of this paper and should be of interest to both the RL and the IV communities.

---

### Official Review · AnonReviewer4 · 2020-11-02
**Promising results, concerns with the assumption, theoretical justifcations, and experiments**

**Rating:** 5
**Confidence:** 3

**Review:**

Summary:

This paper proposes the deep feature instrumental variable algorithm that has two stages of regressions and uses deep features. This paper demonstrates that this algorithm has a good performance on several applications.

This paper is generally well written. Moreover, the experimental results look promising. However, this paper does not theoretically justify its solution that is in general relevant in the IV literature. Instead, this paper emphasizes its algorithm's prediction performance.  It needs some clarification and justification on why the results in this paper can help people better identify and correctly estimate causal effects, which is the primary goal of IV (inference is important in IV). Also, there seem to have some issues with the assumption.

Major comments:

1. Identification: The purpose of IV is to identify and consistently estimate the causal effect in the presence of the unobserved confounders (Angrist et al., 1996). How can we uniquely identify the treatment effect using the DFIV algorithm? And why is the causal effect estimated from the DFIV algorithm consistent? As a reference, causal effects estimated from 2SLS are consistent and asymptotically normal. Moreover, Theorem 2 in Bennett et al (2019) shows the estimated parameters (and treatment effect) converge in probability to the true value. Since IV is not generally used for prediction problems, a careful discussion on identification (and consistency) will be helpful.
2. Assumption 1: In 2SLS, Assumption 1 ($\mathbb{E}[\varepsilon|Z]=0$) is sufficient to prove the estimated treatment effect is consistent and asymptotic normal because the model is linear. However, this assumption is not sufficient for the DFIV algorithm because $\phi(Z)$ is nonlinear. It seems the correct assumption is if $\phi(\cdot)$ is unknown, then $\varepsilon$ needs to be independent of $Z$; if $\phi(\cdot)$ is known, then $\mathbb{E}[\varepsilon|\phi(Z)]=0$.
3. Off-policy policy evaluation experiment:  First, what is the treatment variable in the problem? Second, why does this application use absolute error while the demand and dsprites application use out-of-sample MSE? Third and most importantly, the IVs used in this paper are $Z=(s,a)$.  Why is $\mathbb{E}[\varepsilon|Z]=0$ or $\varepsilon$ independent of $Z$ given $\varepsilon$ is also a function of $s$ and $a$ because in the two expectations in $\varepsilon$, $s^\prime  \sim P(\cdot|s, a)$ and $r\sim R(\cdot|s,a)$?
4. dSprites experiments: The outcome variable $Y$ is quite important when people use IV, but it is not stated in the main text that seems to be confusing. What is the interpretation of $Y$ based on the DGP provided in Appendix D.2? Why should we care about this $Y$?

Minor comments:

1. Typo: P2. six to the last line: ". an unobserved confounder" -> ". $Z$ is an unobserved confounder".
P2. last line: "which we assume to be a continuous" -> "which we assume to be continuous."

---

> ### Author Response · Authors · 2020-11-19
> **[Manuscript Updated] Thank you for the constructive feedback.**
>
> Thank you for the constructive feedback. We updated the manuscript according to your feedback.
>
> 1.  **Identification:**
>     In terms of identification, Newey \& Powell (Econometrica, 2003) have proposed sufficient conditions ensuring identifiability of the structural function, and we have now included this citation, in reference to these conditions, in the revised version of the paper (this reference and conditions were also mentioned in the DeepIV (Hartford et al., 2017) and DeepGMM (Bennett et al., 2019) papers).
>
>     It is beyond the scope of this paper to present consistency results for DFIV. Consistency results for Kernel IV have been established in (Singh et al., NeurIPS 2019) under the assumption that the true structural function lives in the corresponding RKHS. DFIV can be thought of as a method for adaptively learning the feature space itself, based on the data. This makes it more challenging to establish convergence, and a generalization of the results of (Singh et al., 2019) to this case remains an open question.
>
>     It is true that Theorem 2 of Bennett et al. (2019) shows that the estimated parameters converge in probability ot the true value (if the optimization algorithm finds an approximate equilibrium of the game). However, the theorem relies on strong assumptions which do not hold for most neural networks. For example, if we consider hierarchical models, the identification assumption (Assumption 1 in their paper) will typically not hold, since there would be many parameters expressing the same model. Furthermore, the assumption of vanishing Rademacher complexity (Assumption 2) will not be satisfied for many neural networks: see ICLR 2017 paper ``Understanding deep learning requires rethinking generalization'', which states that for neural networks the Rademacher complexity is close to 1, regardless of sample size.
>
> 2. **Assumption 1**:
>     The assumption on $\varepsilon$ adopted here is the same as the one adopted in the DeepIV paper (Hartford et al. 2017) and the DeepGMM paper (Bennett et al., 2019), which both state that $\mathbb{E}[\varepsilon|Z]=0$ is sufficient if we consider the additive confounder $\varepsilon$. As mentioned earlier, identifiability of the structural function under similar assumptions has been discussed in Newey \& Powell (Econometrica, 2003).
>
> 3. **Off-policy policy evaluation experiment**:
>     In the OPE example, the "treatment" variable is $X=(s,a,s',a')$ while the "instrumental" variable is $Z=(s,a)$ as indicated after equation (10). The goal of OPE is to estimate the structural function $Q^\pi(s, a)$ and then use this function to compute the policy value defined as the expectation w.r.t. the initial distribution $\mathbb{E}_{s\sim\rho_0, a\sim\pi} [Q^\pi(s,a)]$. Therefore we measure the accuracy in terms of the error w.r.t. this expectation, in contrast to MSE on out-of-sample data as in other experiments.
> The variables $\varepsilon$ and $Z$ are not independent but uncorrelated as required. Indeed, we have $\mathbb{E}[\varepsilon|Z]=\mathbb{E}[\varepsilon|(s,a)]=0$ where the expectation is w.r.t. $s',a',r$.
>
> 4. **dSprites experiments**:
>     Thank you for the suggestion. We have added the outcome to the main text. There is no intuitive interpretation of this outcome, we just wanted to show here that DFIV can handle complex confounding and high-dimensional treatment.

---

> > ### Comment · AnonReviewer3 · 2020-11-21
> > **Quick comment on identifiability**
> >
> > In addition to the additive noise assumption, I believe Newey requires the completeness assumption to guarantee identifiability in additive(separable) models.

---

> > > ### Author Response · Authors · 2020-11-23
> > > **Regarding to completeness assumption**
> > >
> > > Thank you for pointing this out. The completeness assumption guarantees indeed the existence of a unique solution to the integral equation appearing in IV (see Proposition 2.1 in Newey & Powell for sufficient and necessary conditions for this assumption to hold). This is what we were implicitly referring as the "necessary and sufficient conditions". We have made this explicit in the paper.

---

### Decision · Program_Chairs · 2021-01-07
**Final Decision**

**Decision:**

Accept (Poster)

**Comment:**

The reviewers appreciated the paper's applied neural net approach to the problem of designing features for 2SLS regression for IV analysis as an alternative to sieve approaches. The paper would make a good contribution to ICLR. While the paper does not focus on theory -- learning data-driven features appears to be mostly heuristic -- it should still be grounded in a sound approach to the IV problem, and the reviewers recommend various important technical clarifications regarding the foundations of IV models; the authors should implement these suggestions very carefully and correctly in future versions. For example, even if the structural models are well-specified in that Eq. (5) holds for some parameters, since the dependence is non-linear on parameters, it is not clear when we should expect this to be identifying of (theta_X,theta_Z) (these are in fact not identifiable in general) and when we should expect the proposed method to be consistent.